



# The Thiem team – Adolf and Günther Thiem, two forefathers of hydrogeology

Georg J. Houben[1], Okke Batelaan[2]

[1]Federal Institute for Geosciences and Natural Resources (BGR), Hannover, Germany
[2]National Centre for Groundwater Research and Training (NCGRT), College of Science & Engineering, Flinders University, Adelaide, SA 5001, Australia

*Correspondence to*: Georg Houben (Georg.Houben@bgr.de)

**Abstract.** Adolf and Günther Thiem, father and son, left behind a methodological legacy that many current hydrogeologists are probably unaware of. It goes much beyond the Dupuit-Thiem analytical model for pump test analysis, which is connected to their name. Methods, which we use on a day-to-day basis today, such as isopotential maps, tracer tests and vertical wells were amongst the many contributions which the Thiems either developed or improved. Remarkably, this was not done in a university context but rather as a by-product of their practical work designing and building water supply schemes in countries all over Europe. Some of these water works are still active. Both Thiems were also great science communicators. Their contributions were read and applied in many countries, especially in the US, through a personal connection between Günther and O.E. Meinzer, the leading USGS hydrogeologist of the time.

Keywords

Adolf Thiem, Günther Thiem, O.E. Meinzer, history, pumping test, well construction

## 1 Introduction

The name Thiem appears in many hydrogeological textbooks, most often in the context of the Dupuit-Thiem method, an analytical model for the evaluation of steady-state pumping tests. Few hydrogeologists, however, are aware that there were two engineers of this name, father and son, Adolf and Günther Thiem. Both contributed much more to the current hydrogeological methods than just a somewhat outdated pumping test model. Their work laid the foundations for a diversity of applications and methods still being used today, e.g., tracer tests, well construction, and isopotential maps, and was widely acknowledged even on the international scale, especially in the US. They also planned and supervised the construction of many groundwater supply schemes in several European countries, some of which are still active today, although in modernized form. The focus of this study is thus to investigate the scientific biography of both Thiems and how their contributions found their way into the international canon of methods.





## 2 Adolf Thiem

### 2.1 Biography

Adolf Thiem was born on February 21, 1836, under the full name of Friedrich Wilhelm Adolf Thiem in the town of Liegnitz (now Legnica, Poland) in the Prussian province of Silesia, where he obtained his high school diploma (Herfried Apel, pers. comm.; Anonymous, 1906). His family had been living in Liegnitz at least since the 18th Century. His father was the eponymous Friedrich Wilhelm Adolf Thiem (born 1804), who married Johanna Natalie Julianne Thiem, nee Küpper in 1835. The family was of a craftsman background, but all were self-employed; the father was a master plumber, the grandfather Gottlieb Wilhelm a master nail smith, and the great-great-grandfather Johann a master cartwright. Adolf had a younger brother, Paul Thiem (born 1841 in Liegnitz, died 1883 in Munich), who also became an engineer. Adolf left his parents' house at the age of 14 for apprenticeship and self-study (Vieweg, 1959). He never attended a university but became an autodidact. At the age of 25, he published his first paper in the influential Journal für Gasbeleuchtung (Thiem, 1861), where he introduces himself as an inspector at the gas works of his hometown Liegnitz, a job he still held at least into the following year (Thiem, 1862). In his 1863 paper in the same journal, he signs as inspector of the gas works of the much larger town of Munich (Thiem, 1863), a job he kept until 1865. The early papers already show his mathematical proficiency and his will to improve technical concepts (Thiem, 1861, 1864, 1866). Through contact with Nicolaus Schilling (1826-1894), founder of the now renamed Journal für Gasbeleuchtung und Wasserversorgung (Journal for gas lighting and water supply, based in Munich), he was recommended to Heinrich Gruner (1833-1906), a German engineer based in Basel, Switzerland, at that time. Gruner had mainly built gas works so far but wanted to expand into the water supply market and hired Thiem as an assistant in 1865 (Mommsen, 1962). Gruner introduced the aspiring Thiem to some fundamental French literature, including the works by Henry Darcy (1856) and Jules Dupuit (1854, 1863). His first work assignments led Thiem to the French town of Beaucourt, near Belfort, where he built spring captures and pipelines, and to Winterthur, Switzerland. After a bumpy start, Thiem proved to be an excellent technician, and in 1868 Gruner made him his partner and head of the branch office in Dresden (Mommsen, 1962). The company was called "Heinrich Gruner & Thiem, Ingen. und Unternehmer von Wasseranlagen" (Engineers and entrepreneurs of water schemes). Thiem was mainly tasked with obtaining a share of the quickly expanding market for water supply in Germany. Again, after a bumpy start, Thiem managed to acquire several contracts, mainly convincing his clients through his technical competence. One of the projects was for the historic mining town of Freiberg, Saxony, where he installed a dual system in 1871, consisting of separate spring-fed drinking water and a service water network (Grahn, 1883, 1902). Gruner, however, was not equally happy since Thiem showed much less enthusiasm for financial issues and the day-to-day supervision of the construction sites than for the technical details. Therefore, he decided to move to Dresden himself in 1873 to regain control (Mommsen, 1962). Together, they designed and built the water supply schemes for the cities of Zwickau (1875) and Regensburg (1875), both fed by springs. For the latter, they relocated their company to this town in 1874. In the newspaper announcements from this time, Thiem is mentioned as "Ingenieur von Kamburg, Sachsen-Meiningen" (engineer from the city of Kamburg, Duchy Saxony-Meiningen), where he must have lived briefly. The Regensburg scheme was a technical challenge



since it involved capturing springs located in a river bed, which needed to be protected from the river water itself. Additionally,
the pipeline had to be laid through the bed of the Danube and Regen Rivers, which they accomplished by the intensive use of
divers (Thiem, 1877; Mommsen, 1962). It was not unusual that such projects were financed by issuing stock for a designated
public water supply company, in this case with a value of 1,028,400 German Mark, of which the Gruner & Thiem company
assumed a substantial share of 340,000 Mark (Grahn 1902). The project was so time-consuming that Thiem moved his family
to Regensburg. He had married Luisa Thekla Groß (born 1852 in Zöblitz, died 1931 in Leipzig) in 1871 in Freiberg, while
working there. All his three children were born in Regensburg: Paul Adolf (1874-1907), Ernst Gerhard Günther (1875-1959)
and Katharina Else (1876-?) (Mommsen, 1962; Hoffmann, 2017). Gruner and Thiem´s first truly groundwater-based supply
system was the one for the city of Augsburg (1873-1879). Groundwater head observations for this study were already plotted
in the form of an isopotential map.

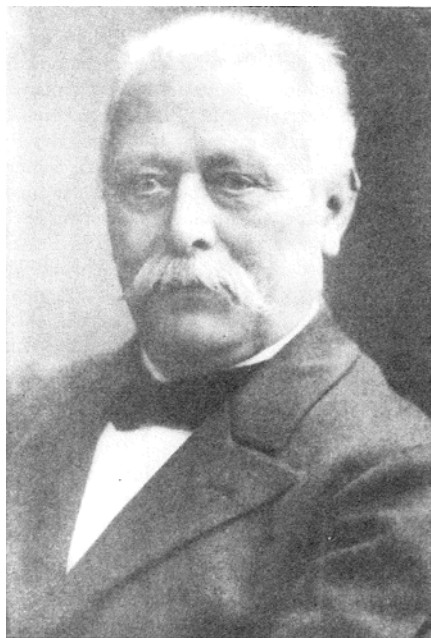


**Figure 1: Adolf Thiem around the year 1900 (Franke and Kleinschroth, 1991).**


Pumping tests, using observation wells to investigate the aquifer response, were already performed by the German engineer
Bernhard Salbach (1833-1894) in Halle, Germany, in 1866 (Houben, 2019). Thiem´s significant improvement, first applied in
Augsburg, was the comparison of the drawdown to predictions by the Dupuit-Thiem model, which he had published previously
(Thiem, 1870, see below). This was probably the first pumping test subjected to a rigorous mathematical evaluation. Another
pumping test in Strassburg, Alsace, received more attention since its results were published in much more detail (Thiem,
1876b). Through their work in Augsburg and Strassburg, Thiem had clearly set the standard for identifying and quantifying





groundwater resources. But he also considered the basic engineering problems of water supply, e.g., the design of pipeline
networks (Thiem, 1876a, 1883a, 1884a, 1885b,c, 1915).

The conflicts between Gruner and Thiem had not abated. Thiem considered himself the underappreciated and underpaid
workhorse, and in 1876 the partnership was dissolved (Mommsen, 1962). Both, independently of each other, moved to Munich,
where several concepts for a central water supply were being considered. Thiem favoured groundwater, based on an intensive
investigation in the fluvial Gleisenthal aquifer and published a detailed report (Thiem, 1878). In the end, the city council
selected a concept proposed by Bernhard Salbach, based on karst springs located 38 km away in the Alps, due to their high
yield, pristine water quality, and the fact that the system was purely gravitational. This proved to be a wise decision since the
system is still the backbone of the city´s water supply today. After the split from Gruner, Thiem successfully promoted himself
by advertising the projects with Grunner as his own exploits. An irate Gruner felt obliged to publish a piece in a Munich
newspaper, denouncing Thiem as a mere assistant, whose responsibility had been to travel, acquire projects, take
measurements, and prepare calculations, which then had to be submitted to Gruner (Gruner 1876).

In 1886, following an invitation by the city mayor Otto Georgi, Thiem moved for the last time to Leipzig. In the first year,
they lived in the Kramerstraße but then moved into the newly built "Haus Pommer" at Hillerstraße Nr. 9 in 1887, which was
to become the Thiem family residence at least until the late 1950s. His consulting company, which at the turn of the century
figured as "A. Thiem & Söhne, Civilingenieure" (A. Thiem & sons, Civil engineers, Mommsen, 1962) became so successful
that he had to rent a separate office already in 1891, located at the Thomaskirchhof 18, right in the city centre, which he later
moved to Quaistraße 2 in 1902 (today Carl-Maria von Webern-Straße). The company employed up to twelve people, including
his two sons. His older son Paul Adolf, a graduated civil and mechanical engineer died in December 1907, aged only 33, a few
months before Adolf (Anonymous, 1908). Adolf Thiem was the leading planner of the groundwater supply scheme for several
larger cities (Table 1), including his new hometown Leipzig, which was expanded in several stages (Thiem, 1881, 1906, 1908).












**Table 1: Main water supplies planned and built by Adolf Thiem (English names in parentheses)**

| Name of city | Name of city today | Comment | References |
|---|---|---|---|
| Freiberg | | 1871, with Gruner | Grahn (1883, 1902) |
| Zwickau | | with Gruner | Grahn 1902, Mommsen (1962) |
| Regensburg | | with Gruner | Thiem (1877a) |
| Augsburg | | with Gruner | Gruner and Thiem (1874), Mommsen (1962) |
| Strassburg | Strasbourg, France | then Germany | Thiem (1876) |
| München (Munich) | | not built | Thiem (1877b, 1880d, 1914) |
| Nürnberg (Nuremberg), also Fürth | | | Thiem (1879a, 1881a) |
| Riga | | then Russia, today Latvia | Salm (1893), Thiem (1883b, 1888e) |
| Leipzig | | | Thiem (1881b,e,e 1883c, 1885a, 1889, 1890a, 1890b, 1891, 1892a, 1892b, 1906, 1908, 1911) |
| Gera | | | Thiem (1884c), Grahn (1902) |
| Stralsund | | not built | Thiem (1888d) |
| Malmö, Sweden | | 1890, with J.G. Richert | Svensson (2013) |
| Potsdam | | | Thiem (1892d) |
| Charlottenburg | | now part of Berlin | Thiem (1897a, 1913a) |
| Mainz | | incl. Laubenheim | Thiem (1897b,c), Grahn (1902) |



| Dessau | | 1897 | Grahn (1902), Pfeffer (1906) |
|---|---|---|---|
| Breslau | Wrocław, Poland | then Germany | Anonymous (1902) |
| Prag (Prague) | Praha, Czech Republic | then Austria-Hungary | Anonymous (1903) |
| Braunschweig (Brunswik) | | | Grahn (1902), von Feilitzsch (1904) |
| Waldenburg | Wałbrzych, PL | then Germany | Lummert (1905) |
| Landeshut | Kamienna Góra, PL | then Germany | Thiem (1909) |


Other cities in Germany he was working for include – in alphabetical order - Biebrich, Blasewitz, Crimmitschau, Eilenburg,
Essen, Frankenstein (Ząbkowice Śląskie, PL), Greifswald, Harburg/Hamburg, Hirschberg (Jelenia Góra, PL), Hohenstein,
Kiel, Liegnitz (Legnica, PL), Limbach, Magdeburg (Thiem & Fränkel 1902), Mansfeld, Markranstädt, Meerane, Metz,
Mittweida, Oels (Oleśnica, PL), Plauen, Posen (Poznan, PL), Warmbrunn (Cieplice Śląskie-Zdrój, PL), Wismar and Zeitz
(Grahn 1902; Anonymous, 1906; Dyck, 1986). His expertise was also valued abroad (Anonymous, 1906, 1952; Dyck, 1986)
and, additional to the entries in Table 1, led him to work in Romania (Bucharest, Czernowitz, Klausenburg/Cluj-Napoca),
Scandinavia (Åbo/Turku, Finland, Malmö (Sweden), and Porto Alegre (Brazil). His work was not restricted to studies of
aquifers and wells but also encompassed the hydraulics of pipeline networks, the improvement of pumps, the development of
water treatment techniques (especially iron removal) and even the construction of water towers, e.g. the still existing tower in
Strasbourg from 1878, the first with a semi-spherical wrought iron tank (Thiem, 1876, 1877a, 1878, 1880c, 1883a, 1884a,
1885b, 1896, 1897a, 1894b, 1898b, 1915, 1929q; Grahn and Thiem, 1885). He briefly worked on inland navigation, in
particular on the hauling of cargo vessels, on the Hohensaaten-Spandau canal near Berlin, work that was presented in a
conference in Paris in 1892 (Thiem 1892c). Curiously, his home base is given as Eberswalde. He offered his clients the full
package, ranging from groundwater exploration to planning and construction of wells and pipeline networks, water treatment
plants and storage tanks, including economic considerations (Thiem, 1884b). He was probably one of the first to use the term
"sustainability" (Nachhaltigkeit) in the context of groundwater (Thiem, 1881a). He had observed the groundwater levels in
observation wells located along the Leipzig-Grimma train track over the course of 15 years. The relatively stable drawdowns
led him to the conclusion that the drawdown caused by the extraction for the Leipzig water supply had become stable and
extraction was thus sustainable (Thiem, 1881a).
In 1892, Thiem received the honorary title of "Königlich Sächsischer Baurat" (Royal Saxonian building officer). Probably in
1899, he received the "Königlich Sächsischer Verdienst-Orden" (Royal Saxonian Order of Merit), as of 1900 he proudly added
the title "Ritter 2c" (knight, second class) to his entry in the Leipzig address book. A striking feature of his work ethic was that
he never took out any patent, in order to foster the advancement of science (Anonymous, 1906, 1952). When asked about it by





his pupils, he would smile and answer "Dies ist für die Allgemeinheit und nicht für mich alleine da. (This is for the public and
not for me alone)" (Anonymous 1952). However, this claim is not entirely true since he took at least one patent on a water
valve that would automatically close after a sudden pressure loss, e.g. caused by a pipeline rupture (Thiem, 1894a). On his
70th birthday, he was honoured by a page-long biographical sketch in the Journal für Gasbeleuchtung und Wasserversorgung,
which states his role as founding father of hydro(geo)logy (Anonymous, 1906). Adolf Thiem died after short but severe
suffering at the age of 72 in Leipzig on May 2, 1908 (Anonymous, 1908). He was buried there on the Südfriedhof in an
honorary grave that still exists (Fig. 2). It is only a few meters away from the still active Probstheida water works and its
impressive water tower, which Thiem designed shortly before his death. In 1912, the city of Leipzig named a street after him
(Thiemstraße), which still bears this name today and leads to the Probstheida water works (Fig. 2). Several important German
hydrologists such as Emil Prinz, Max Rother and his son, Günther Thiem were his pupils. In Germany, his legacy was
recognized and kept alive, evidenced by several commemorative articles (Thiem, 1929q; Prinz, 1936; Anonymous 1949;
Anonymous 1952; Anonymous 1958; Vieweg 1958a,b, 1959; Dyck, 1986; Engemann, 1989). As late as 1956, his seminal
1870 publication was reprinted (Thiem 1956).

Thiem´s contributions to the growing field of hydrogeology were also noted outside Germany, already during his lifetime. His
work for the water supply of Leipzig was considered important enough to be presented at the world exhibition in Chicago in
1893 (Hillger, 1893). In their 1899 book on groundwater flow, Franklin Hiram King and Charles Sumner Slichter cite seven
of A. Thiem´s papers, including those on tracer tests and other German papers by Lueger and Hagen (King and Slichter, 1899).

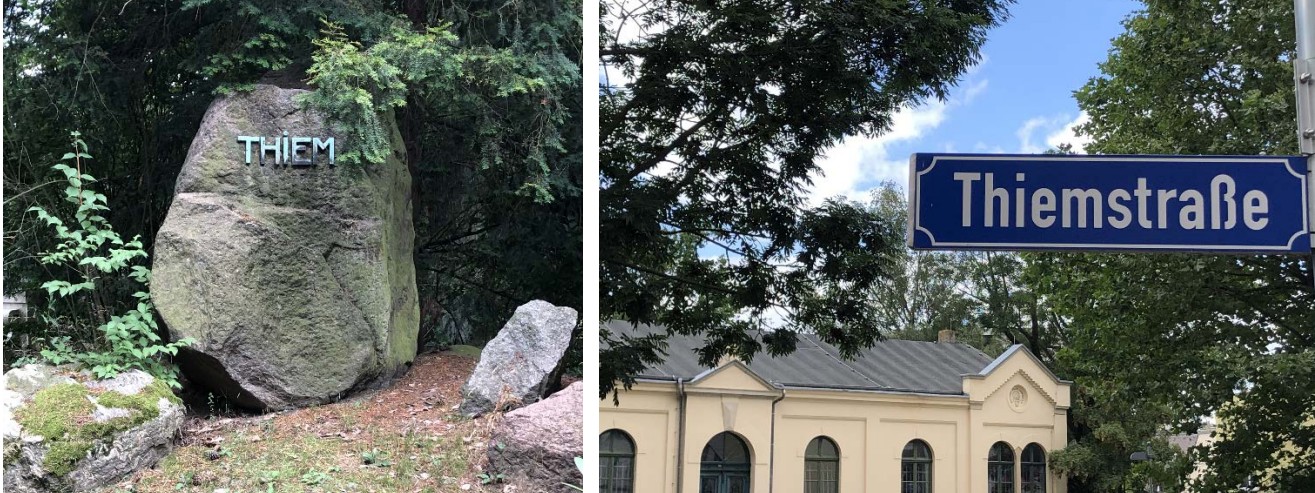

**Figure 2: (left) Grave of honour of the Thiem family at the Südfriedhof, Leipzig. The gravestone is an erratic block found during**
**the construction of the neighbouring monument (Völkerschlachtdenkmal) commemorating the decisive Battle of Leipzig against**
**Napoleon 1813, (right) road sign of the Thiemstraße (Straße = street) in Leipzig (Photos: Houben).**






## 2.2 Contributions to pumping tests

The analytical model describing the radial flow of groundwater to a well embedded in a horizontal circular island aquifer is
sometimes called the Dupuit model after Jules Dupuit (1863), sometimes the Thiem model after Adolf Thiem (1870) or
Günther Thiem (1906), and sometimes the Dupuit-Thiem model. It is therefore important to compare the seminal contributions.
After analysis of open-channel flow, in chapter VIII of his 1863 publication, Dupuit turned his attention to flow in permeable
soil (Du mouvement de l´eau travers les terrains perméables). Based on his work on open channel flow, Dupuit stated that the
slope of a groundwater table should follow a parabolic equation of the type of Equation 1:
$\quad i = \alpha \cdot Q + \beta \cdot Q^2$ $\hspace{10cm}$ (1)

Where i is the slope, Q is the flow rate, and alpha and beta are coefficients. This is basically identical to the later Forchheimer
equation (Forchheimer, 1901). However, Dupuit realized that the velocity term $\beta \cdot Q^2$ could be ignored due to the commonly
very low flow velocities of groundwater. Assuming a radial symmetry and a horizontal aquifer, he then derived the fundamental
equations describing groundwater flow to a well at steady state, for both water table and artesian aquifers. Thiem (1870) had
clearly read Dupuit´s paper, as he duly cites it and ends his paper with a literal quote in French from Dupuit. Günther Thiem
claimed that his father had actually been a friend of Dupuit, which is technically possible since Dupuit died in 1876, well after
the 1870 publication by Adolf (Thiem, 1951). There is, however, no other evidence that both knew each other, apart from
Günther´s claim. Thiem´s paper follows parts of the outline of Dupuit´s chapter VIII closely. So, was Thiem, just a copycat?
Not quite! In his equations, Dupuit used two heights of the water table above the impermeable aquitard, (1) $h_0$ in the well itself
at the well radius $r_0$ (la hauteur de l´eau dans le puit) and (2) H at the outer radius of the cone of influence (la hauteur de l´eau
extérieure) at a radius R (le rayon du massif filtrant). While the choice of these two points was sufficient for the mathematical
derivation, they both were a rather poor choice from a practical point of view. The water levels in the well were often affected
by additional, non-laminar head losses caused by the well tubing itself, something which Dupuit was aware of (see below) but
chose to ignore. He also gave no practical hints on how to obtain the outer limit. He only realized that the value for the outer
radius is of limited influence as it appears in a logarithmic term $(\log(R/r_0))$. As such, the equations were of limited practical
use and were not taken up by practitioners.

It was Adolf Thiem´s merit to have grounded the Dupuit equation in the real world. He used two observation wells located
within the cone of depression at different radii $r_1$ and $r_2$, thus avoiding the problems of turbulent losses in the well and of
finding the radius of influence. While Dupuit (1863) takes precedence for the mathematical derivation (Ritzi and Bobeck,
2008), Thiem (1870) and his later papers (e.g. Thiem, 1876b) converted the method into a practical tool and popularized it. It





is thus justified to call the method the Dupuit-Thiem model. Remarkably, his first-ever paper on groundwater became a classic.
For the confined case, it takes the form of Equation 2:
$$h_2 - h_1 = \frac{Q}{2 \cdot \pi \cdot K_{aq} \cdot B} \cdot \ln\left(\frac{r_2}{r_1}\right)$$    (2)
with
$h_1, h_2$    =    head at radial distance $r_1$, $r_2$ [L]
$Q$    =    pumping rate [L³/T]
$K_{aq}$    =    hydraulic conductivity [L/T]
$B$    =    constant thickness of confined aquifer [L]
$r_1, r_2$    =    radius from well axis, with $r_1 < r_2$ [L]

Although the first well-documented pumping test in Germany was performed in 1866 in Beesen near Halle/Saale by Bernhard
Salbach (Houben, 2019), Adolf Thiem's work defined some of the standard procedures. Already for his first pumping tests in
Augsburg, Strassburg/Alsace und Munich he developed several approaches that are still in use today (Thiem, 1876b, 1879a,
1880). To delineate the geometry of the cone of depression and the radius of influence, he installed several observation wells,
both perpendicular and parallel to the estimated flow direction of groundwater (Fig. 3). For this purpose, he mostly used
Abyssinian wells ("Norton tubes"), sturdy prefabricated well tubes, usually of 50 mm inner diameter, which could be rammed
into the ground and recovered – if necessary – afterwards. They were spaced more closely near to the well and further apart
from it (Fig. 3). He also insisted on installing observation wells outside of the radius of influence to study the influence of
natural variations of the groundwater levels, e.g., the ones caused by varying river water levels. By default, not only the
drawdown phases for different pumping rates (Fig. 3) but also the recovery phase was observed (Thiem, 1876b). Another
regular procedure was measuring the groundwater temperature during the test and taking water samples for later analysis.
Already in Strassburg 1874/5 he used a "Locomobile mit Centrifugalpumpe", a submerged centrifugal pump driven by an
external steam engine (Thiem, 1876b). The drive shaft of the pump was probably connected to the engine via a belt, like a
primitive drive shaft pump.

Adolf Thiem used one procedure, which is not common anymore: he increased the depth of the pumping well during the test
to find productive zones, as he had realized early on that thin layers of high conductivity provide a disproportional yield of
water. He was also probably the first to notice – and quantify - the difference between horizontal and vertical hydraulic
conductivity. From the results of his pumping test in Strassburg, he determined a value of eight for the ratio of horizontal to
vertical conductivity (Thiem, 1876b). This is remarkably similar to the default value of ten recommended in most textbooks
today. During his exploration of the hydrogeology around Leipzig, Thiem realized the concept of multi-aquifer systems, i.e.
the presence of several aquifers stacked on top of each other, separated by aquitards (Thiem, 1881a). He referred to these
individual aquifers as "Grundwasseretagen" (groundwater floors/levels).

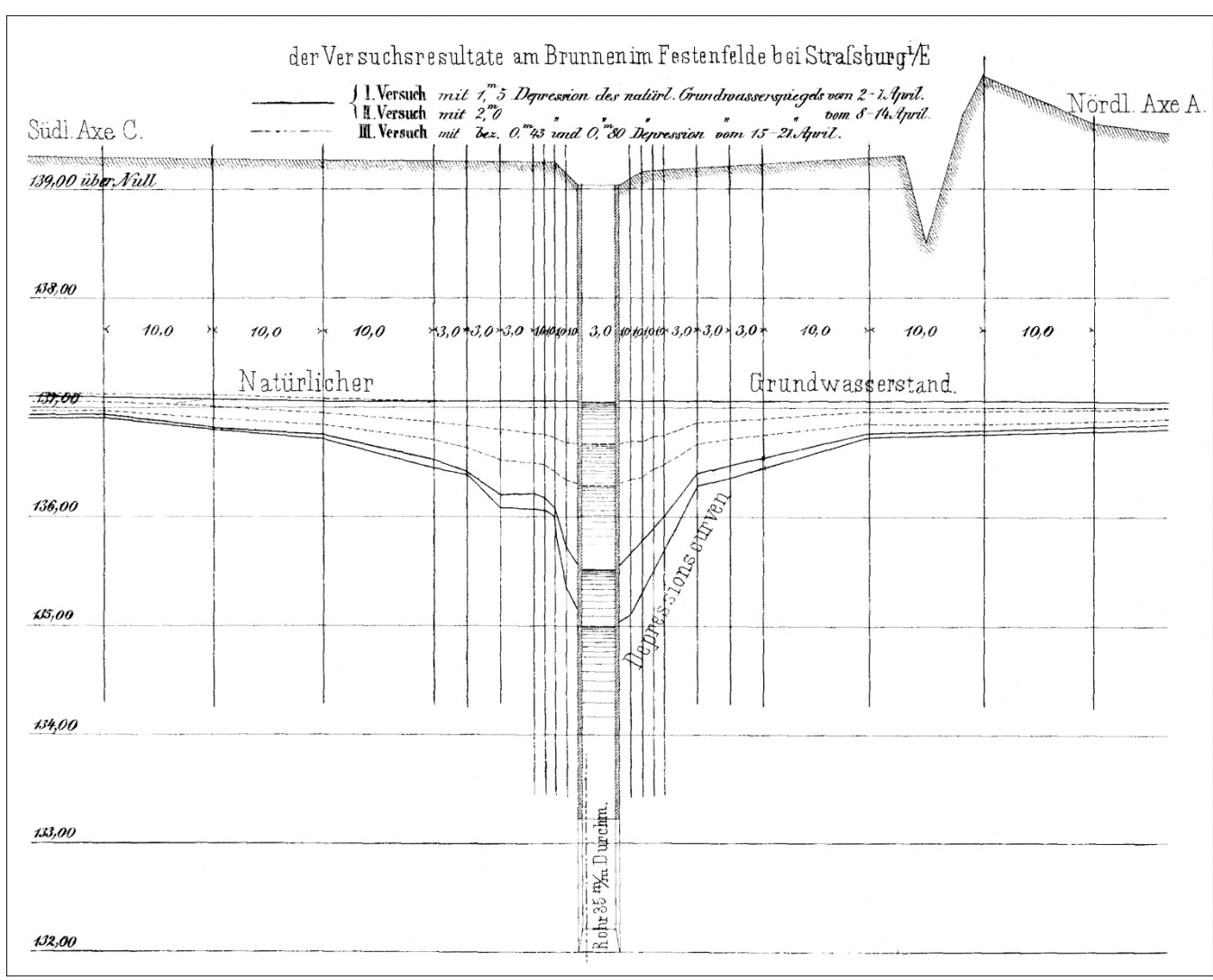


**Figure 3: Sketch of the cones of depression for different flow rates obtained during pumping tests in Festenfeld near**
**Strassburg/Alsace. The vertical black lines are the logarithmically (10 m, 3 m, 1m spacing) arranged observation wells**
**(Thiem, 1876b). Translation: Natürlicher Grundwasserstand = natural groundwater level, Rohr = well diameter: 35**
**mm, Depressionskurven = pumping level curves, Versuch = Test, Versuchsresultate = test results, über Null = above**
**(French) sea level, Südl./Nördl. Axe = southern/northern axis.**

Dupuit (1863) had realized that flow in pipes connected to the well, e.g. a riser pipe, can cause additional head losses. To
address this, he brought back a velocity term from his studies on pipe flow and added it as a second term, very similar to the
one shown in Equation 3. Again, Thiem (1870) follows him in this, adding a velocity term in the slightly different form of the





well-known Darcy-Weisbach equation (Eq. 3). Interestingly, Dupuit (1863) references his previous work as the source for the
velocity term, and Thiem (1870) calls it a "well-known equation" without citing any reference. Both thus ignore the
contribution by Julius Weisbach (1845).

$$h_2 - h_1 = \frac{Q}{2 \cdot \pi \cdot K_{aq} \cdot B} \cdot \ln\left(\frac{r_2}{r_1}\right) + f_D \cdot \frac{L_B}{4 \cdot r_b^5 \cdot \pi^2 \cdot g} \cdot Q^2 \qquad (3)$$
with
$f_D$     =     Darcy friction coefficient
$L_B$     =     length of borehole (L)
$r_b$     =     radius of casing/screen (L)
$g$     =     acceleration of gravity (L/T²)

Dupuit (1863) realized that he could use the velocity term to investigate the relative influence of pipe flow on well hydraulics.
He retroactively studied two wells in Grenelle and Passy, both near Paris. Again, Thiem (1870, 1879) converted Dupuit´s
theoretical approach into a practical tool, the step-discharge test, which is still being used today. Therefore, he simplified
Equation 3 to

$$H - h = K_{aq} \cdot + k_w \cdot Q^2 \qquad (4)$$
with
$H$     =     head in well at zero flow [L]
$h$     =     head in well while pumping (steady state) [L]
$Q$     =     pumping rate [L³/T]
$K_{aq}$     =     hydraulic conductivity [L/T]
$k_w$     =     well loss coefficient [T²/L⁵]

This equation is still the main model to interpret step-discharge tests today. In his pump tests, Thiem plotted the drawdown s
as a function of different pumping rates Q and could identify the presence and quantify the contribution of the velocity term,
or in other words, the non-linear laminar and turbulent losses of the well itself (Houben, 2015a, b). If the s-Q pairs plotted on
a straight line, the flow was laminar and the velocity term negligible. Any deviations from a straight line could then be
attributed to additional well losses and quantified. Therefore, Thiem usually employed several pumping rates during his tests,
plotted the resulting drawdown curves and evaluated the contribution of non-laminar flow (e.g. Thiem, 1876a,b, 1879, 1880).





Adolf Thiem realized that removing fines from the aquifer at high pumping rates can improve its hydraulic conductivity and
thereby discovered the principle of well development (Thiem, 1875). In some cases, he took this to the limit and beyond. In
the course of a pumping test in Strassburg/Alsace, the highest pumping rate of 136 l/s (490 m³/h) induced such a high degree
of suffusion that the ground around the well subsided, and the well tubing was deformed (Thiem, 1875).

The method for pumping test evaluation after Adolf Thiem (1870) remained one of the most important hydrogeological tools
for several decades. It was intensively discussed and applied in the USA (Wenzel, 1932, 1933, 1936, 1942; Meinzer, 1934),
which can be traced back to the good contacts of Günther Thiem to the leading USGS hydrogeologist of its times, Oscar
Meinzer (see section 4). The Dupuit-Thiem method was not without flaws: as a steady state method, it commonly required
long times until the drawdown had become stable and needed two observation wells. The transient method by Theis (1935),
which does not require steady drawdown and can do with one observation well, was the first serious challenger but remained
problematic due to the use of type curves, which was both tedious and a bit subjective. Only its later simplification by Cooper
und Jacob (1947) relegated the Dupuit-Thiem method to the second place.

Nevertheless, the Dupuit-Thiem equation can still be found in many textbooks (e.g. Batu, 1998; Kruseman and de Ridder,
2000; Bear, 2007; Kresic, 2007; Kasenow, 2010). Due to its geometrical set-up and simple mathematics, it is often used to
teach students how to derive analytical models for groundwater flow (e.g. Hendriks, 2010). It is still helpful for the design of
water wells and the planning of construction dewatering (Houben 2015a,b). For pumping tests, it has become a niche method
when steady-state pumping test data are available (Misstear, 2001). The Dupuit-Thiem equation forms the basis for several
later analytical models, including the old but still commonly used Forchheimer (1901) model, which describes the contribution
of non-linear flow processes in the flow towards wells (Houben, 2015a,b). The Forchheimer equation consists of two terms;
the first is the Dupuit-Thiem equation, which describes the linear laminar losses. The second term describes the non-linear
laminar losses. Until today, the Dupuit-Thiem equation is used as a base-case for validation or as quality control for more
advanced analytical models (see Tügel et al., 2016 for examples). Despite its simplicity and biblical age of 150 years, to this
day, the Dupuit-Thiem equation is still an important method for groundwater professionals worldwide.

Prior to the full development of vertical wells, many hydrologists used backfilled drainage trenches instead, which could be of
substantial length and depth (Houben, 2019). While working for the water supply of Winterthur, Switzerland, with Heinrich
Gruner, Adolf Thiem considered such an option (Thiem, 1870). Therefore, he adapted his equation for well flow to a linear
sink. Despite its simplicity, it only considered the height of the water table from the constant-head boundary to the drain in a
2D projection (Thiem, 1870). This was probably the first model for horizontal wells.





## 2.3 Contributions to well design

The first pumping wells Thiem had used were shaft wells of large diameter, e.g. in Strassburg. They were difficult and expensive to build and often displayed poor performance. He realized that he could overcome these problems by developing the concept of the Norton wells (Abyssinian wells) further, which he had used as observation wells during his pump tests. In 1881-83, for the water works of Naunhof (Leipzig), he increased their diameter to 150 mm, which still allowed them to be rammed into the subsurface. At first, he tried to emulate the shaft wells by installing so-called "Ringbrunnen" (ring wells), a central collector shaft surrounded by up to 20 individual rammed vertical wells, aligned on a circle with a radius of 10 m from the shaft (Engemann, 1989). The vertical wells were drilled first and then partially excavated down to the depth of the pipeline towards the central collector (Fig. 4). The latter still proved to be a difficult and expensive construction, and the many wells tended to interfere with each other. The Ringbrunnen were operated until 1926 (Engemann, 1989).

Later, he installed vertical well galleries, connected to a central siphon pipeline. This concept proved to be much more useful and cost-effective and became the standard. However, the vertical wells caused a lot of trouble due to corrosion, sand intake and incrustations, which often led to their complete failure to deliver water after only a few years. Thiem even equipped his wells with a noose, attached to the bottom, which could be used to pull out the whole well (Fig. 5). Later, a detachable screen was tried (Thiem, 1925). Thiem introduced cast iron as a material for screen and casing, which was more corrosion-resistant than the forged iron used before. Since the slots in the cast or forged iron screens were – due to technical reasons - quite wide (often up to 1 cm), sand control was a critical problem. Many wells filled with sand eroded from the aquifer quite quickly. The solution used by Thiem was to wrap fine metal meshes around the screens, which, however, were prone to blockage by the very fines they were supposed to retain and by corrosion and incrustations. Due to their small diameter and the described clogging processes, the yield of the early Thiem wells was quite small, often in the range of a few cubic meters per hour. Therefore, Thiem had to install 225 of them for the first well field of Leipzig in 1883 and 300 for a later one (1907) in the same town (Thiem, 1925). Thiem kept tinkering with the well design, e.g. by simplifying the design (Fig. 5), increasing the diameter to 150 mm (1907 in Leipzig), installing rubber seals and introducing copper pipes, which were lighter, easier to manufacture and much more corrosion-resistant, although more expensive. For the Nuremberg water works, the tedious and problematic metal meshes were replaced by an artificial gravel pack, a technique that had already been used for horizontal drains (Thiem, 1879; Houben, 2019). In Nuremberg, Thiem (1879) proposed a gravel pack of four layers with gradually increasing grain size towards the well (2, 4, 8, 15 mm). The well itself was made from perforated brickwork.



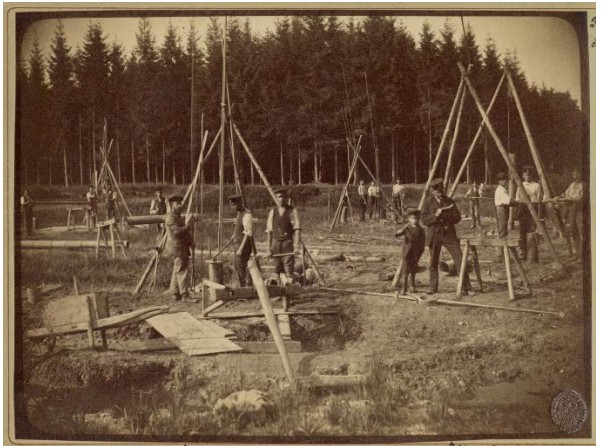

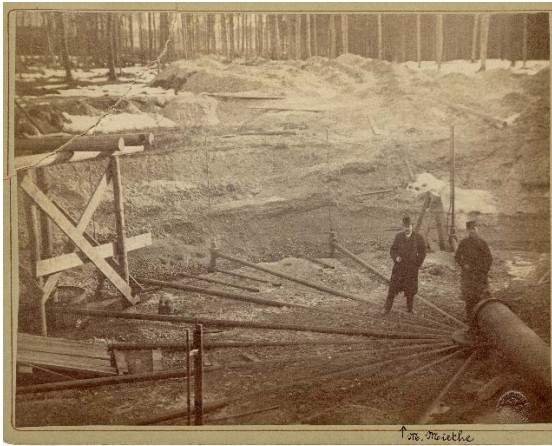

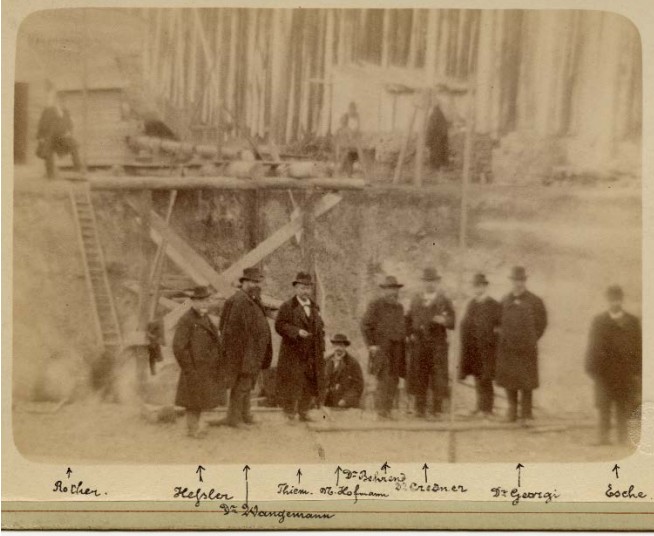

**Figure 4: Construction of a "Ringbrunnen" at Leipzig-Naunhof, around 1887. Upper left: drilling of vertical wells, upper right: view of the radial pipelines connecting the vertical wells (visible at the end) to the central collector shaft, lower left: Adolf Thiem visiting the construction site (third from left, lower row). Also present is Prof. Credner, head of the Saxonian Geological Survey and Max Rother (left), one of Thiem´s pupils (Photos: Stadtarchiv Leipzig).**





**Figure 5: Well designs by Adolf Thiem used in Leipzig. Left: first design from 1883, cast iron screen, riser pipe from wrought iron; right: simplified design from 1894, backflow valve omitted, suction pipe now made of copper (Thiem, 1925).**

Thiem also found time to study the flow of groundwater towards wells under laboratory conditions. In 1879 and 1882, Gustav Oesten had presented sandtank experiments on the groundwater flow to vertical, partially penetrating wells installed at two different depths in a square box (Oesten, 1879a, 1882a,b,c). Using colour tracers, he correctly observed that the highest flow velocities occurred around the screen. For a short screen installed at shallow depth, he found that coloured water from the bottom of the aquifer did not flow to the well (Oesten, 1882a). He thus postulated an interface separating a pumping-affected from a not affected area. Only a deeper placement of the screen induced flow from below. Adolf Thiem was very unhappy with this and stated in his rebuttal that his previous theoretical work had already clarified how water should flow around a well (Thiem, 1879d, 1882). However, he still felt obliged to perform his own sandtank experiments, which he called "*demonstratio*





*ad oculos*" (Latin for "demonstration to the eyes"). At first, he used a square box but later a wedge-shaped sand body to
simulate the convergent flow towards the well. The main objection of Thiem on the experiment of Oesten (1882a) was that Mr
Oesten infiltrated water through a small trench at the surface of the box. As this did not represent the reality of flow to wells,
Thiem allowed water to be infiltrated from one side over the entire thickness of the sand and the water level in this reservoir
was kept constant by an overflow (basically a constant head boundary). The well was simulated by a little sieve body from
which water was extracted. The images indicate that the bottom of the well was probably not closed. The well screen only
covered the uppermost third of the saturated aquifer thickness. The flow paths were visualized by injecting small volumes of
coloured water at different depths at the inflow side. This conclusively showed that water from below the screened interval
also entered the well, inducing a vertical flow component close to the well and elevated inflow rates at both the top and the
bottom of the screen. Thus, Thiem had conclusively demonstrated the flow field around a partially penetrating well. Mr. Oesten
responded to the rebuttal (Oesten, 1882b), claiming rather unconvincingly that Thiem had not sufficiently considered the
influence of capillarity, but the case was settled.

Unbeknownst to many well designers, Adolf Thiem defined one of the most critical and most criticized values, the maximum
permissible entrance velocity. Many textbooks and international standards on well design cite a value of 0.03 m/s (0.1 ft/s),
e.g. Campbell and Lehr (1973), Driscoll (1986), Sterrett (2007). Keeping the entrance velocity below this value is said to curb
head losses, maintain fully laminar flow conditions, prevent suffusion of sand particles, minimize incrustation build-up and
even to control corrosion. The value is sometimes attributed to Bennison (1947), who, however, presented neither theoretical
concepts nor experimental or field data to back up his claim. It is very likely that this value goes back to experiments executed
by Adolph Thiem, while he was designing wells and their gravel packs for the Nuremberg waterworks (Thiem, 1879). Thiem
instinctively understood that the flow velocity of groundwater is the critical parameter that controls particle mobilisation and
thus sand intake. Therefore, he investigated the minimum vertical flow velocity required to keep grains of different diameters
in suspension. At velocities below, the grains would not be transported. For sand grains up to a grain diameter of 0.25 mm he
obtained maximum flow velocities under which no transport would take place of 0.028 m/s, which is basically the
recommended value above. The value found its way into the influential German textbooks by Smreker (1914) and Thiem´s
pupil Emil Prinz (1919) and the monograph by G. Thiem (1928). It is quite probable that US hydrologists became aware of
this value from the German literature and through personal exchanges between Oscar Meinzer of the USGS and Günther Thiem
(see below) and adopted it without further questioning.

For the water supply of the town of Greifswald, located at the German Baltic Coast, Adolf Thiem built a rather unusual
construction in 1890 to extract groundwater. He had found an artesian aquifer of 6 m thickness under a confining layer of 5 m
of glacial till (Houben, 2019). Instead of wells, he had a trench of 9 m depth and 450 m length constructed, equipped with two
strings of perforated stoneware tubes of 500 mm diameter each, installed at different depths and then backfilled. He also had





an impervious underground cut-off wall installed to impound the groundwater, allowing it to flow towards the town by gravity
alone. Unfortunately, this most likely very expensive construction never lived up to the expectations. The yield was very low
at 10.8 m³/h and soon had to be augmented by additional vertical wells.
**2.4 Development of tracer test methods**
Although reports on - sometimes involuntary - tracer experiments in karst aquifers predate the 19[th] century, Adolf Thiem
played a crucial role in developing tracer experiments into a scientific instrument, especially for porous aquifers (Thiem, 1887,
1888). His first field tests were done in 1886 in the towns of Greifswald and Stralsund, located at the Baltic Coast of Germany.
He dissolved 75 to 100 kg of table salt (NaCl) in water and measured the breakthrough curves in several observation wells
(Thiem, 1888). Therefore, the chloride concentrations were determined via titration with silver nitrate, using potassium
chromate as an indicator. During a tracer test in Plauen (Saxony), he observed five to six tracer peaks, which he attributed to
the heterogeneity of the aquifer. To understand the fundamental processes of tracer migration Thiem (1888) performed
laboratory experiments using a sand column of 4 m length. Based on his experiences, Thiem (1888) was the first to postulate
fundamental requirements for tracer chemicals: (1) non-reactive, (2) non-toxic, (3) cheap and (4) easy and quantitative analysis.
**2.5 Equipotential and hydrogeological maps**
During his work in Augsburg with Gruner, Adolf Thiem made extensive use of Norton (or Abyssinian) wells, small but thick-
walled pipe screens that could be rammed into the ground, to measure groundwater levels. Since they also determined the
ground elevation of the observation wells, they were able to construct one of the world´s earliest isopotential maps in 1873
(Mommsen, 1962; Dassargues et al., 2021). Strangely enough, Thiem considered the map produced for a later project in
Strassburg, Alsace (now Strasbourg, France) as his first isopotential map, probably because he published a detailed account of
this study in the Journal für Gasbeleuchtung und Wasserversorgung (Thiem, 1876b), which was widely received and
acclaimed. Figure 6 shows a typical example of Thiem´s clear graphical style, showing equipotentials based on observation
wells, time series of groundwater levels and cross-sections showing aquifer thickness and water table.

Thiem immediately realized the influence of the water level of the neighbouring river Rhine on groundwater levels and thus
constructed two equipotential maps, one for high and one for low river stages (Thiem, 1878). Due to its importance, the original
drawing of the equipotential map was donated to the German Museum (Deutsches Museum) in Munich (Thiem, 1929q, 1941a),
the most important technical collection of Germany. Unfortunately, it seems to have been lost during the war, as a request for
it with the museum archive in 2020 by the authors led to no results. However, a copy is reproduced in some publications of
Günther Thiem (1929q, 1931f, 1941a).



**Figure 6: Isopotential map from the Leipzig-Naunhof study (Thiem, 1881). Blue isopotentials are from 1880, the red ones for 1881. Black dots and numbers show the observations wells. The straight black line in the west is a train track, and the shaded areas are villages.**

Mainly due to the increasing demand for mineral resources, geological mapping became an important task in Germany during the second half of the 19[th] century. The role of unconsolidated rocks as aquifers, however, was not overlooked. Adolf Thiem contributed a chapter "On the hydrology of the old river bed of the River Mulde near Naunhof" to the "Annotations on the Geological Map of the Kingdom of Saxony, section Naunhof", Sheet 27 (near Leipzig), one of the first hydrogeological



contributions to a geological map (Thiem 1881c; Sauer 1881; Sauer et al., 1906). The cooperation with geologists was thus no
anathema to Thiem. It had been the geologist Prof. Hermann Credner (Fig. 4), head of the Saxonian Geological Survey, who
pointed Thiem towards Naunhof, where the second water works for Leipzig was installed in 1887, the largest and most modern
groundwater works of Europe at the time (Credner 1883; Thiem 1892a,b; Heinker 2005). Credner later supported Günther
Thiem when he wanted to become a member of the German Geological Society in 1911.

**2.6 Artificial groundwater recharge**

Thiem quickly realized that not all aquifers were productive enough to satisfy the demand and that an augmentation via surface
water might be useful (Thiem, 1898). Early on, he studied bank filtration, e.g. in Fürth in 1880 and for the town of Essen, and
recommended using temperature as a tracer to distinguish ground and surface water (Thiem, 1898). He was also aware of the
danger of colmation of the riverbed (Thiem, 1929q). For the water supply of Stralsund, Thiem had unsuccessfully proposed
artificial groundwater recharge via drainage trenches (Thiem, 1888b), a concept already applied in Chemnitz in 1875, using
trenches with an artificial sand bed (see discussion in Thiem, 1898; Houben, 2019). However, Thiem´s Swedish pupil Johann
Gustaf Richert (1857-1934) perfected the concept (Svensson, 2013). It was implemented for the first time in Göteborg in 1898.
Richert published his experiences in a book in German (Richert, 1911), and the concept became quite popular in Germany
after the turn of the century, especially in the Ruhr valley.

**2.7 Construction dewatering**

The construction of deep basements often requires working in the saturated zone and thus the control of groundwater. In the
19$^{th}$ century, this problem was – if not avoided altogether – tackled by encapsulating the construction site and sealing it off
from the surrounding groundwater, e.g. by ramming sheet piles, injecting cement or freezing parts of the aquifer. These
procedures were technically demanding, costly and not always successful. Adolf Thiem realized that dewatering by verticals
wells was a viable alternative since the well type he had developed could be installed cheaply and quickly, and his equations
allowed him to dimension the dewatering scheme. In 1886, Thiem applied this concept, using a shaft well, for the first time in
the construction of the Leipzig water supply in Naunhof (Prinz, 1907; Thiem, 1929q, 1931f). Therefore, Thiem can be
considered one of the founding fathers of construction dewatering.

**2.7 Scientific feuds**

Thiem regularly attended conferences, e.g. those of the German Association of Water Professionals (DVGW), and was an avid
contributor to the discussions (e.g. Thiem 1880b,c,d, 1885c, 188b,c). He did not shy away from voicing controversial opinions,
which led to some prolonged scientific feuds.

The main opponent of Adolf Thiem was Oskar Smreker, born in 1854 on Castle Görzhof/Cilli, Austria-Hungary (now Celje,
Slovenia), and died in Paris in 1935. He was a graduate of the Swiss Technical University (ETH) Zurich (1870-1874), where





he, much later, in 1914, at the age of 60, received his PhD on a groundwater-related study (Smreker, 1914a). In 1876, he was
hired by Heinrich Gruner in Regensburg as a replacement for A. Thiem, after Gruner and Thiem had parted ways, but he was
sacked in 1877 (Mommsen, 1962). After several years as an engineer in Germany and Italy, Smreker founded a successful
company in Mannheim, Germany, in 1882 that designed and built many groundwater supply systems in Germany and abroad.
Smreker published several papers (Smreker, 1878, 1879, 1881, 1883, 1907), criticising both the work of Darcy (1856) and
Thiem (1870, 1876b). He doubted the validity of the Darcy law - and the Dupuit-Thiem equation deducted from it - due to the
supposed ignorance of the increase of velocity around a well. He even formulated his own non-linear law of groundwater
movement and dared to use the results of Thiem´s pumping tests from Strassburg to test it (Smreker, 1878). Adolf Thiem
responded by citing ample literature based on both field and experimental data, which showed the validity of Darcy´s law for
practically all applications (Thiem, 1880).

Even after Thiem had died in 1908, Smreker would not relent. In his 1914 PhD thesis, several papers, and his textbook, Smreker
still attacks the validity of Darcy's law and upholds his alternative law (Smreker, 1914a,b, 1915a,b,c,d,e). He argued that "*The*
*Darcy law […] fails completely when applied to the principle of groundwater abstraction, because the differences in velocities*
*at the varying distances from the well are large*" (Smreker, 1914). Several prominent authors, including Max Rother (1855-
1928), Adolf Thiem´s last collaborator, felt obliged to publish a defence of the Darcy law. In the middle of the First World
War (WW1) and shortly afterwards, a war of papers ensued across several journals and countries and arguments flew back
(Brix, 1915; Rother, 1915, 1916a,b, 1919a,b, 1920; Lummert, 1916a,b, 1917a,b; Hocheder 1919) and forth (Smreker,
1915a,b,c,d,e,f, 1916a,b, 1918, 1919, 1920a,b,c), with Smreker receiving support from Hache (1919) and Henneberg (1919).
Based on an extensive experimental comparison of equations using a Darcy permeameter, which he calls "Thiem apparatus",
Krüger (1918) found the best fit using a modified Smreker equation. Other authors, like Weyrauch (1916), the Dutchman J.
Versluys (1915, 1919), the Austrian-Hungarian J. Zavadil (1915) and Zunker (1920), tried to reconcile the approaches by
investigating their limits. The latter also proposed a new equation based on experimental data. In 1919, the Journal für
Gasbeleuchtung und Wasserversorgung apparently had enough of the discussion and tried to declare it finished (Anonymous
1919), to no avail (Rother 1920; Smreker 1920a,b,c). Adolf´s successor, his son Günther Thiem participated only marginally
in the feud (Thiem 1920i,l). He probably did not want to compromise his role as neutral editor of his journal (3.5). The feud
lost steam in the early 1920s, after more than 40 years of struggle. Although several review papers had tried to declare
Smreker´s approach to be the correct one (Krüger 1918; Hache 1919), his struggle was in vain, and his equation fell into
oblivion and is hardly cited today (Benedikt et al., 2018). Unbeknownst to most participants of the feud, Philipp Forchheimer,
who was only marginally involved in it (Lummert 1916b), had already solved the problem in 1901 by proposing the law today
known as Forchheimer law (Forchheimer 1901). It expands the Darcy law with a velocity term that can be used when flow
velocities are high, e.g. in the vicinity of pumping wells. This fixes the deficiency of the Darcy law that Smreker had correctly
identified. With low velocities, the Forchheimer equation reduces to the Darcy law, which thus remains valid for most
situations. Smreker´s feud with the Thiem School must have been quite bitter, as Smreker does not mention any hydraulic





study of neither Adolf nor Günther Thiem in his otherwise excellent book (Smreker, 1914). This is quite unusual for a time
when there were few published studies available, and Thiem had already been recognized as the founding father of
hydrogeology in Germany.

Another hydrologist who got into trouble with Adolf Thiem was Gustav Oesten, a civil engineer and sub-director of the Berlin
water works, later the author of an influential textbook on water supply that went through several editions (Oesten, 1904). He
had published on the flow of groundwater to well screens based on sandtank experiments and interpreted them in a non-Darcian
manner (Oesten, 1879a), which Thiem attacked in a quite sarcastic style (Thiem, 1879c; Oesten, 1879b). In 1882, Oesten
published basically the same results in a different journal (Oesten, 1882a). Again, Thiem attacked his interpretations and even
conducted experiments to show his point (Thiem, 1882; Oesten, 1882b). Details can be found in section 2.3.
**3 Günther Thiem**
**3.1 Biography**
Günther Thiem was born under the full name Ernst Gerhard Günther Thiem on October 11, 1875, in Regensburg, Bavaria,
where his father was working with Heinrich Gruner (1833-1906) at that time (Fig. 7). After his father had relocated to Leipzig
in 1886, he attended the renowned Thomasschule, Germany´s oldest public school, founded in 1212, which was right next
door to his childhood home in the Hillerstraße. He started his academic career in 1895, studying philosophy at the University
of Leipzig. In 1896 he changed to civil engineering at the Königlich Technische Hochschule (Royal Technical University) in
Stuttgart to follow the classes of Robert Weyrauch (1874-1924) and Otto Lueger (1843-1911), the last being Germany´s
leading expert on water supply and author of influential textbooks (Lueger, 1883, 1895). During the semester breaks, Günther
worked in his father´s consulting company. Lueger in his book "The water supply of towns" (Lueger, 1895), advocated for the
use of springs and groundwater instead of surface water (Loehnert, 2013). However, some of his theoretical concepts were
wrong; he followed the doctrine that groundwater under free water table conditions could not flow upwards (de Vries, 2006).
In 1901 he reappeared in Leipzig with the title "Regierungs-Bauführer" (government building headman), which indicates that
he intended to join the saxonian state administration. But this was not meant to be. Instead, he pursued his PhD in Stuttgart
(section 3.2) and later took over the family consulting company after the rather sudden deaths of his older brother in 1907 and
his father in 1908 (section 3.4).

He married Erna Carola Auguste Goelitz (1887-1976) in Marburg in 1909. They had three children, all born in Leipzig:
Auguste Luisa Ingeborg (born 1911), Anna Else Erika (born 1913), and Karl Wolf Gunther (1917-2015), the latter a renowned
art historian and head of the graphical collection of the state art gallery in Stuttgart (Hoffmann, 2017; Herfried Apel, pers.
comm). After the death of Adolf Thiem, Günther´s family moved into the old Thiem residence at Hillerstraße 9, where they
stayed at least until 1949 (entry in the last available address book) but probably even longer until Günther´s death and possibly



beyond. Adolf´s widow Thekla moved to the neighbouring Schwägrichenstraße, where she lived until her death in 1931. In
the address book, she appears with the description "Privata", indicating a rich widow who could live from her inherited means.

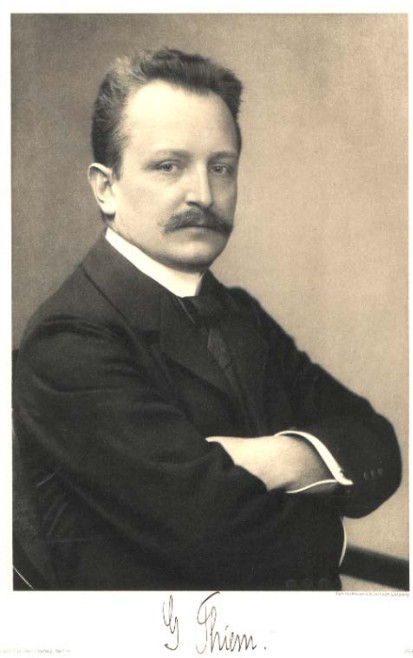 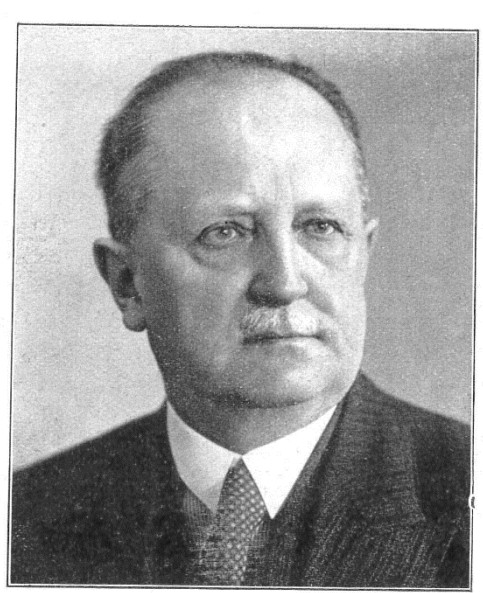

**Figure 7: Photos of Günther Thiem. Left: around 1910 (Anonymous 1910), right around 1940 (Thiem, 1941a).**


### 3.2 Thiem (1906) PhD thesis

Otto Lueger was also the advisor of Thiem´s PhD thesis, which Günther dedicated to his father (Thiem, 1906). It was probably
one of the first PhD studies solely dedicated to groundwater and was widely received in Germany and abroad. The PhD was
remarkably short; 45 pages with 3 annexes, providing 10 borehole descriptions, 3 tables with results of calculations and 8 plans
or cross-sections. The thesis had no formal reference list but referred in the text to publications of six authors (Darcy, A. Thiem,
Slichter, Forchheimer, Dupuit, Lueger). Verbatim quotes were referenced from Slichter and Dupuit in respectively English
and French. In the thesis, he presented the so-called ε-Verfahren (epsilon method). In essence, it was the Dupuit-Thiem
pumptest analysis method for obtaining the hydraulic conductivity. However, instead of using a hydraulic conductivity K, he
defined ε, which he called the unit capacity, as the product of the K and a unit cross-section normal to the groundwater flow.
He derived and presented for unconfined and confined flow to wells, equations for ε, i.e., the Dupuit-Thiem equations. He then
applied this method by performing ten pump tests to estimate the groundwater flow in a 6 km long section of the Iser River
valley (now Jizera River) near its confluence with the Elbe River, close to the city of Altbunzlau (now Stará Boleslav, Czech
Republic). The pumping tests were part of a study to develop groundwater resources for Prague, a project initiated by Adolf
Thiem (Anonymous, 1903). The field investigation was executed in the first half of 1902. He showed that the Iser River is the





receptor of the groundwater flow and that the higher the river bed is above the base of the unconfined aquifer and the closer
one is to the river, the more vertical upward flow there is, which was in contradiction to the ideas of his advisor Lueger (1895).

The last chapter of Thiem's thesis is probably one of the first published extensive analyses of groundwater-surface water
interaction. Thiem explained and presented in clear figures how equipotential lines are differently oriented towards a river
dependent on gaining or losing river conditions (Fig. 8). But also he showed how during an infiltrating flood wave passing
through the river the equipotential lines change of curvature near the river. Hence, he recognized and described the process of
bank infiltration and storage. During 5 months, in support of studying groundwater-surface water interaction, he observed
groundwater levels in piezometers at different distances from the river at the ten pump test locations. In one of the ten locations,
he suffered data loss due to vandalism of his piezometer, apparently an issue of all times. By calculating the changing gradients,
he observed, e.g. on March 25, 1902, that the high river water levels caused infiltrating conditions in the valley aquifer. Based
on observed strongly changing gradients in the time frame of 48 hours, he concluded that groundwater level observations
during at least one year are required to obtain an average gradient with which the groundwater flow to the river can be
estimated. He also extensively discussed the temporal changes in groundwater-surface water interaction and sources of
extracted water under the influence of seasonal groundwater level variations and the regime of a near-river located well. In
designing the well field, Thiem aimed to avoid extracting low-quality surface water. Hence, Thiem developed an analytical
equation to estimate the required distance between the river and the well, based on phreatic flow between two assumed fully
penetrating canals (representing the river and the well). In the same chapter, he discussed the different infiltration and recharge
characteristics of the study area; low on the loamy valley soils and high on the sandy terraces. Moreover, he described the
strongly delayed response of rainfall on the groundwater levels, warning that the delay is generally well underestimated.

The proposed ε-Verfahren never became widely popular under this name, despite being discussed in detail in the book by Prinz
(1919) in German but also in French by Imbeaux (1921, 1930). The approach Günther Thiem proposed was actually not novel
as his father essentially already published in 1870 the derivation of the Dupuit-Thiem equation for estimating the hydraulic
conductivity. Nevertheless, the 1906 PhD thesis very clearly details and applies the method and is well cited, at least 521 times
(Google Scholar, Oct 2020). Often the thesis is erroneously cited as the source of the Dupuit-Thiem model or Thiem equation
(e.g. Wenzel, 1936; Meinzer and Wenzel, 1940), but this honour belongs to Adolf Thiem (1870), which has so far received
only 30 citations. The clear exposition of the Dupuit-Thiem equation and Günther Thiem's support in transferring his method
to the US (see below) explain the erroneous citation.





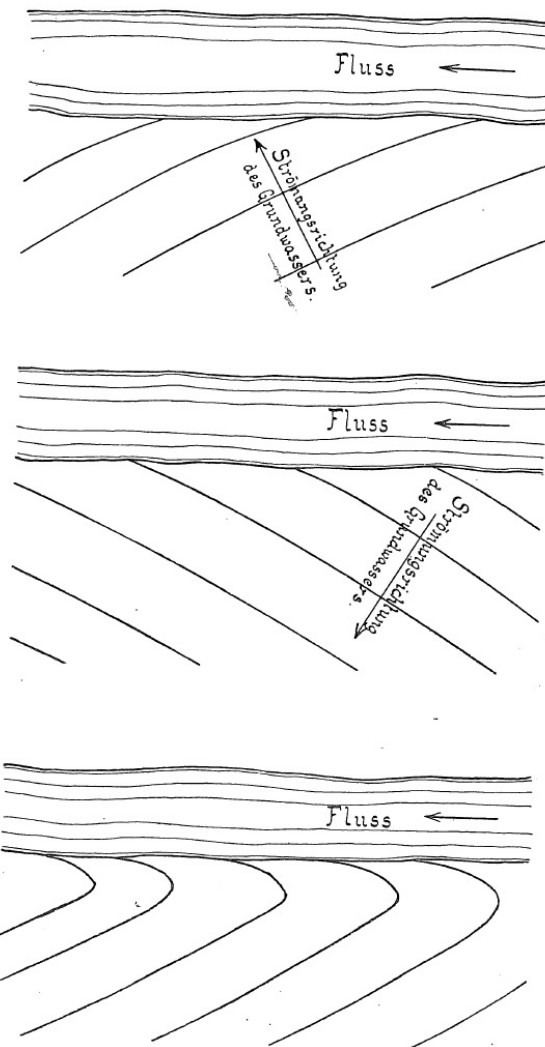


**Figure 8: Groundwater-surface water interaction at Iser River near Prague, top: gaining conditions; middle: losing conditions; bottom: bank infiltration during river flood conditions (Thiem, 1906). Translation : Fluss = river, Strömungsrichtung des Grundwassers = flow direction of groundwater**


### 3.3 Work overseas

After graduating in 1900, Günther Thiem went to the US and worked in New York for the Hering & Fuller consulting company. One of the founders was the famous civil engineer Rudolph Hering (1847-1923), member of the "Hall of Fame" of the American Water Works Association and eponym of the "Rudolph Hering Medal", awarded by the American Society of Civil Engineers for outstanding contributions to environmental engineering. Being of German descent, Hering had been sent by his parents to Dresden to attend school and university. Whether he came into contact with Adolf Thiem during this period remains





unclear. One of Günther Thiem´s projects in the US was building the water supply for the city of Jersey, New Jersey. He also
travelled to Egypt, India and Ceylon (Sri Lanka) during this time (Thiem 1915c, 1936c, 1955a). In 1903, he returned to Leipzig
and became a junior partner in his father´s company. While the bulk of the work there was in Germany, he was also involved
in projects in Austria-Hungary, Switzerland and Russia (details see below).

**3.4 Consulting Engineer**

After the death of his older brother and father, Günther took over the consulting company in Leipzig in 1908, employing five
to seven engineers and several technical staff (Anonymous 1910). In 1911, he moved the offices to Marschnerstraße 13, in
1915 to Plagwitzer Straße 9 and finally in 1939 to Plagwitzer Straße 7 (today Käthe-Kollwitz-Straße), which was basically in
the same corner house as his home in Hillerstraße 9. All mentioned buildings survived the war with minor damage, were nicely
refurbished after the reunification and still exist today (Fig. 9). Public water supply companies were his main clients. For them,
he designed and supervised the construction of many water supply schemes in Germany and abroad (Table 2). Most of them
were based on groundwater and a few on bank filtration, which he considered artificial groundwater (Thiem 1919k). He also
served in the city council of Leipzig (1913-1918 and 1921-1922). In 1912 he was appointed as "Gerichtlicher
Sachverständiger" (surveyor appointed by the court). During the First World War, he served in the German Army as field
engineer and published papers on military aspects, e.g. the construction and drainage of trenches (Thiem, 1915a, 1916e, 1917e),
field water supply (Thiem, 1917a, 1919c) and the disinfection of water (Thiem, 1916d, 1918a, 1918d, 1919c). For his efforts,
he was awarded the Saxonian medal of war merit (Kriegsverdienstkreuz), a fact that is curiously never mentioned in any of
his later biographies (Anonymous 1917).

After the war, he applied his skills in the growing field of lignite mining, which had major impacts on groundwater resources
through the dewatering of the open-pit mines in central Germany and Bavaria (Thiem, 1920b,m, 1921c, 1922a,b, 1923d,
1924a,b, 1928e, 1929b,i, 1930b, 1935b, 1937d, 1938a, 1939e, 1940e, 1952). In his publications of this time, he introduced
himself as "Montanhydrologe" (mining hydrologist) and tried to convince the mining engineers that geohydrology was an
important contribution to their field. The industrial water supply also became important (Thiem, 1919k, 1920k, 1922e,
1924c,d,e,f 1929l, 1931e, 1935d,e, 1937a). Building on the work of his father, he was also an important contributor to the
improvement of the design and construction of vertical wells (Thiem, 1911d, 1916b, 1917d, 1919f, 1920c,d,j, 1923c,f, 1924h,
1925a, 1928a,d, 1929f, 1936a, 1938d, 1941b, 1942, 1951b,c, 1953c,d). Similar to his father, he investigated hydraulic and
economic aspects of pipeline networks (Thiem, 1910b, 1910h, 1912b,c, 1915d, 1918c, 1919b,d,e, 1920a, 1924c,e, 1931b,i,n,
1932a,d,e, 1938b, 1954) and their maintenance (Thiem, 1914b, 1929d). Water treatment, especially the removal of ferrous
iron, was a side issue (Thiem, 1910i, 1914d, 1915b, 1924d, 1928c,f, 1929a, 1931m). He also designed and, unlike his father,
patented technical equipment, amongst them a device to measure groundwater levels (Thiem, 1908), a detachable riser pipe
(Thiem 1911d), a water meter (Thiem 1911e,f 1912a), a device for screened wells allowing the injection of chemical reactants
to dissolve incrustations (Thiem, 1931d), an acid-proof coating for metal well screens (Thiem 1931j), a rubber pipe seal (Thiem





1933d), a check valve with the wonderfully German name "Rückschlagklappenventil" (Thiem, 1935b) and a gate valve
(Thiem, 1937c).

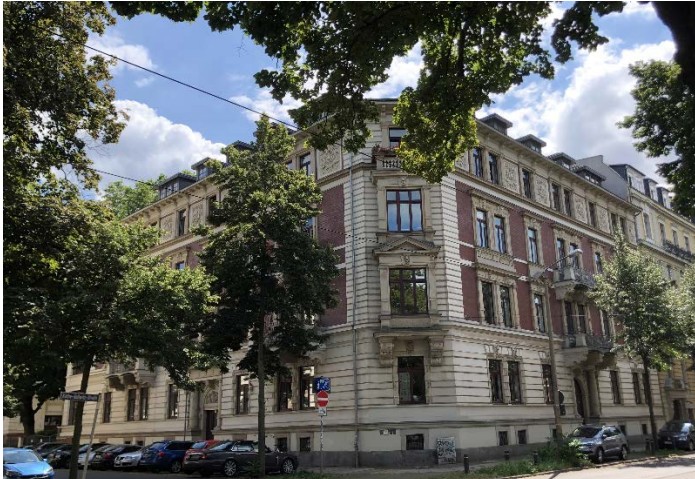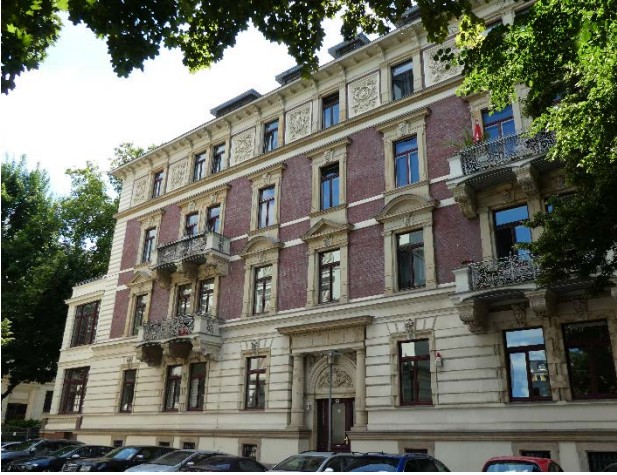

**Figure 9: (left) Corner house Hillerstraße 9 (left) and Plagwitzer Straße 7 (right, today named Käthe-Kollwitz-Straße), Günther had**
**his offices in Plagwitzer Straße 9 (yellow building to the right) since 1915 and finally in Plagwitzer Straße 7 since 1939, (right)**
**Hillerstraße 9, the Thiem family residence: Adolf and his family lived there on the second floor since 1887, Günther took over in**
**1909 (Photos: Houben).**

Due to his age, he did not serve in the Second World War (WW2) but contributed several short publications detailing the water
supply for troops in the field, copying his work produced during WWI (Thiem, 1937b, 1940b).

**Table 2: Main water supplies planned and built by Günther Thiem (English names in parentheses)**

| Name of city | Name of city today | Comment | References |
|---|---|---|---|
| Prag (Prague), Altbunzlau, Czech Republic | Praha, Stará Boleslav | then Austria-Hungary | Thiem (1906) |
| Landeshut | Kamienna Góra, Poland | then Germany | Thiem (1909b) |
| Harburg | | today part of Hamburg | Thiem (1910a) |
| Wilhelmsburg | | today part of Hamburg | Thiem (1910c) |
| Leipzig | | expansion of previous schemes | Thiem (1910d,j, 1911a,b, 1912b, 1914c, |





| | | | 1915d, 1920g, 1922c, 1935a,d, 1957) |
|---|---|---|---|
| Czernowitz | Czernowice, Ukraine | then Romania | Thiem (1910e, 1910f, 1929c,n) |
| Magdeburg | | | Thiem (1910g, 1921b) |
| Mönchengladbach | | | Thiem (1911c) |
| St. Petersburg, Russia | | | Thiem (1913b,c, 1929k,m) |
| Vaasa (Wasa), Finland | | then Russia | Thiem (1913e), Juuti and Katko (2006) |
| Meerane | | | Thiem (1914d) |
| Kempten | | | Thiem (1915e) |
| Aue | | | Thiem (1916c, 1923a) |
| Zeitz | | | Thiem (1919e, 1920f) |
| Danzig | Gdansk, PL | | Thiem (1919a,h,j) |
| Halle | | | Thiem (1919i,l 1921a), Winterer (1919) |
| Mitau, Latvia | Jelgava | | Thiem (1929e,n,o) |
| Posen, Poland | Poznan | | Thiem and Matakiewicz (1923) |
| Zittau | | | Thiem (1929g,h,p) |
| Tampere (Tammerfors), Finland | | | Gagneur and Thiem (1928, 1929) |
| Wolmsdorf | | | Thiem (1930b) |
| Bautzen | | | Thiem (1931b,h,l) |
| Saaz, CZ | Zatec | | Thiem (1932b,c) |
| Reichenberg, CZ | Liberec | | Thiem (1933a,d, 1934a, 1939d) |
| St. Moritz, Switzerland | | | Thiem (1933b,c, 1934b,c) |





| Samaden, CH | | | Thiem (1936b,d) |
|---|---|---|---|
| Dessau | | | Thiem (1955b) |


Other cities he worked for include Zwickau, Freiberg, Spremberg, Gera, Linz (Austria), and Suceava, Romania, then Austria-
Hungary (Pöpel 1956). In his study for Mönchengladbach, he lists the prices for several of his hydrogeological investigations,
including drilling costs and their duration (Thiem 1911c). The investigations in Prague and Leipzig took about 200 days each
and cost 51,000 and 30,000 German Mark. The study in Czernowitz took 67 days, while the one for Mönchengladbach required
150 days, both at the cost of about 15,000 Mark. To roughly convert these prices into Euro, one has to multiply them by 5.2.
During his work in Switzerland in the early 1930s, he briefly became technical director of the Hydrotechnik AG, Zurich (Thiem
1933c).
**3.5 Editor, publisher and author**
In 1914, Günther Thiem became the executive editor of the „Internationale Zeitschrift für Wasser-Versorgung" (International
Journal for Water Supply), founded by the „Internationaler Verband der Wassersachverständigen" (International Association
of Water Experts), the first international journal exclusively dedicated to hydrology. The journal was published through his
own publishing company "Technischer Verlag Dr.-Ing. Günther Thiem". Rudolph Hering (USA), Édouard Imbeaux (France),
Felice Poggi (Italy) and J.G. Richert (Sweden) acted as additional editors (Fig. 10). His contacts thus went further than the US
(see Section 4) and, despite all political problems, included the French-speaking world, e.g. through Prof. Imbeaux, whom he
calls *"..a dear old friend"* in the letter shown in Figure 11. Even in 1916, when the war between Germany and France was in
its third year, Günther Thiem published a paper on the water supply for Nice, France (Thiem, 1916a). The friendship with
Imbeaux outlasted the war, and as early as 1921, Imbeaux promoted the Thiem epsilon method in an article (Imbeaux, 1921).
Contributions to the journal came from all over the world, including from leading US hydrologists of the time, such as Charles
Slichter (Slichter, 1915). Günther also republished several of his father´s older publications (A. Thiem, 1914, 1915, 1918,

637 1920).


Interestingly, the 1917 issue of the journal still mentions all original foreign editors, although Germany was at war with France
and Italy (Hoefer von Heimhalt from Vienna and his former teacher Robert Weyrauch from Stuttgart had been added
meanwhile). The journal was active throughout WW1, but only published articles in German. In 1918, Günther Thiem realized
that the term "International" in both the journal title and the name of the association was awkward at a time of war and dropped
it. The names of Hering, Imbeaux and Poggi disappeared as coeditors, while H. Peter from Zurich, Switzerland, was added. In
mid-1919, the journal was renamed "Zeitschrift für Wasserversorgung und Abwasserkunde" (Journal for Water Supply and
Wastewater Science). In 1920, he decided to give up the journal, and it was subsequently merged into the journal "Wasser und
Gas", which appeared until 1934, with Günther serving as associated editor. He also worked in the same position for the





"Kalender für das Gas- und Wasserfach", a yearbook for the gas and water field, which appeared between 1921 and 1938.
After WW2, Günther Thiem did reappear as editor of a journal: from 1951 to 1956 he was listed as co-worker of the journal
"Bohrtechnik, Brunnenbau (Drilling technique, well construction)". Ironically, after his death in 1959, the East German
government forgot to delete his now inactive publishing company from the public registry. Finally, in 2007, several years after
the German reunification did the authorities finally delete it.




**Figure 10: Header of the "Internationale Zeitschrift für Wasser-Versorgung" (1917), showing the international co-editors and the**
**journal title in different languages.**

Günther Thiem was a prolific author. He left a legacy of around 200 publications treating theoretical concepts, technical
inventions, case studies from his consulting work and promoting the general benefit of groundwater. He repeatedly published
papers or booklets that summarized the gained knowledge on hydrogeology (e.g. Thiem, 1907, 1909a, 1913d, 1914a, 1917b,c,
1918b,e, 1919g, 1920e,h, 1922d, 1923e, 1925b,c, 1926a, 1927a,b, 1928b, 1929j,l, 1930a,c, 1931a,f,g,k, 1939f, 1940d,f, 1941a,
1951a, 1953a,b, 1955c; Thiem and Gagneur, 1929). His interest in international hydrological affairs is evidenced by several
review articles on foreign water supply schemes, stretching as far as the Soviet Union and Egypt (Thiem 1915c, 1916a, 1923b,
1924g, 1936c). Many of his publications appear in a series published by himself, called "Thiems Hydrologische Sammlung"
(Thiem's Hydrological collection), a series of small booklets, which often are reprints of some of his papers published in
Journals. He was also a great communicator whose oral explanations of by integrals supported hydrological calculations, were
even understandable for lawyers (Grahmann, 1960). This was often necessary since the quantitative methods introduced by
both Thiems were initially often met with scepticism. As late as the early 20[th] century, a senior government official told Günther





Thiem "*Your whole hydrology is nonsense, I simply build well after well, until I obtain the desired quantity of water*" (Thiem
1911c). Luckily, these random searches for groundwater, often "aided" by the use of the divining rod, were slowly overcome
due to the persistent work and the publications by both Thiems. During his search for groundwater for the city of Bautzen,
Günther actually hired two water diviners to compare their results to his drill holes, with less than convincing results for the
divining rods (Thiem 193b,h,l).
**3.6 Honours**
Like his father´s work, Günther´s contributions to Leipzig and Prague's water supply were considered important enough to be
shown at the world exhibition in Brussels 1910, where he was even awarded a silver medal (Stoffers, 1910). The occasions of
his 60[th], 75[th] and 80th birthdays in 1935 and 1955 were honoured by the publication of short biographies (Anonymous, 1935,
1950, 1955, 1956; Lang 1950; Paavel 1955; Herzner 1955). Although not of working class background, Thiem was also
honoured by the East German communists, who took over in Leipzig after WW2. In December 1952, they awarded him the
somewhat peculiar title "Verdienter Techniker des Volkes" (merited technician of the people), one of the first to receive this
honour (Henneberg, 1952). In the same year, he was appointed Ehrensenator (honorary senator) of the Hochschule für
Bauwesen (University of Construction) in Leipzig (Schöne, 1959). Not to be outdone, he also received prices from West
Germany. In 1956, the German Association for Gas and Water (DVGW) awarded him their highest honorary price, the Bunsen-
Pettenkofer-Ehrentafel (Ehrentafel = shield of honour, Anonymous 1956), and the Technical University of Stuttgart
commemorated the 50[th] anniversary of his PhD by awarding him the Golden PhD diploma (Pöpel 1956; Schöne, 1956). His
death was mourned in both East and West Germany (Anonymous 1959a; Anonymous 1959b; Schöne 1959; Grahmann 1960).
**4 Günther Thiem and Oscar Edward Meinzer**
The work by Adolf Thiem had already been noted in US literature (e.g. King and Slichter, 1899), but it was Günther who
popularized the Thiem methods abroad, especially in the US. Trying to understand the background to why generally in the US
literature (Ritzi and Bobeck, 2008) the Dupuit-Thiem equation is called the Thiem method after Thiem (1906), and why it
became so popular, we investigated the contacts between Günther Thiem and US scientists, especially Oscar Edward Meinzer.

C.V. Theis, former District Geologist and Division Scientist at the USGS Division of Ground Water from 1930 till his official
'retirement' in 1970, was interviewed by John Bredehoeft in 1985 (Theis, 1985; Bredehoeft, 2008). "CV" was at that time
already 85 years old. Although he took time to respond, his mind was still sharp, and he remembered quite clearly (Bredehoeft,
2008). Bredehoeft asked CV about the pumping test in Grand Island, Nebraska, run by the USGS (Wenzel, 1932, 1933, 1936).
Theis replied that Meinzer had gone to Europe to meet Günther Thiem, who had been using pumping tests for water supply,
and "*brought back the idea and to really try it out*". He said "*it was the only one at that time [in this country], …, well, no,
who was it that presumably made some sort of a pumping test in Pennsylvania?*". He also related that "*this was just before*





*Hitler's time and Meinzer was sending back to Thiem various baskets of food because Thiem was having a hard time there*".
The food baskets were most likely sent after the war since Thiem was a successful businessman before it.

The Grand Island pumping test was planned in 1930 under the supervision of O.E. Meinzer, who was since 1912 Geologist in
charge of the Division of Ground Water of the USGS. The measurements took place in summer 1931; results were described
in short in Wenzel (1932, 1933) and fully documented in Wenzel (1936). The goal of the two performed pumping tests was
"*to ascertain the accuracy of the Thiem method and to investigate the possibilities of determining specific yield by a pumping*
*test*" (Wenzel, 1936). The Wenzel 1932 and 1936 publications both have in their title "The Thiem method for determining
permeability of water-bearing materials…" and described the method extensively. Meinzer (1932) also explained the method,
it is likely that he presented the method already at a meeting of the Society of Economic Geologists in New York City, Dec.
29, 1928: "*Mimeographed copies of the paper in abbreviated form had been sent to the members prior to the meeting. The*
*paper has been revised and enlarged for the present publication*" (Meinzer, 1932). Both Meinzer and Wenzel referred to A.
Thiem, particularly the Thiem (1887) tracer test paper but not to the Thiem (1870) paper. However, Meinzer (1934) referenced
also Adolf Thiem (1870): "*He introduced field methods for making tests of the flow of ground water and applied the laws of*
*flow in developing water supplies. Under his influence Germany became the leading country in supplying the cities with ground*
*water. The results of his work appeared in a number of papers, the first in 1870*". Hence, we may assume that Meinzer was,
since at least 1928, aware of the Thiem method based on Thiem (1906) and Thiem (1870). The Wenzel (1936) Water-Supply
Paper 679A effectively established the Thiem (1906) method as a standard for permeability assessment of pumping tests and
received broad uptake. In the acknowledgement of Wenzel (1936), Leland Wenzel thanks Günther Thiem for his criticism of
the manuscript, which shows the existence of contacts between Thiem and the USGS at least during the 1930's.

It took between 66 and 30 years after respectively Thiem (1870) and Thiem (1906) until the Thiem type of pumping test was
introduced and made popular in the US. Although Meinzer (1925, 1928) realized the importance of compressibility and
elasticity of aquifers in the 1920's, the dominant groundwater flow theory was steady state and dictated by the Dupuit-Thiem
model until Theis published his transient solution in 1935 (Theis, 1935; Deming, 2002). The slow acceptance of the Theis
equation (in part by Meinzer) meant that by 1936 the USGS Water-Supply Paper 679-A still could widely introduce and make
the Thiem method popular in the US.

To investigate in more detail the contacts between Günther Thiem and the USGS, we requested a search of the US National
Archives through record group 57 of the U.S. Geological Survey. This resulted in Entry A1 593, "Correspondence and Other
Records Relating to the International Committee on Underground Water, 1936 – 1946", about 42 pages of relevant
correspondence between Günther Thiem and Oscar Edward Meinzer dated between 1 December 1936 and 23 August 1940
(Thiem and Meinzer, 1936-1940). The correspondence consists of 17 letters from Thiem to Meinzer and one to Dr Fleming,
13 letters from Meinzer to Thiem, one from Dr Fleming to Thiem, one from the Chief Clerk to Thiem, and a copy of a



publication about Thiem 65 years old (Anonymous, 1935). Thiem writes in German to Meinzer, while Meinzer writes back in English. However, it is clear that both have a good command of the other language. Of the 13 letters of Thiem, only three seem to have been translated. The first letter of 1 December 1936 appears to have been translated by Meinzer himself in handwritten notes on the letter of Thiem (Fig. 11). The second and third translated letters are typewritten with the likely purpose of transferring them to a colleague. Some remarks by Thiem concerning the (upcoming) war in Europe receive particular interest and are translated in English on the original letters in Meinzer's handwriting. On the letter of Thiem of 3 November 1939, Meinzer wrote the translation: "*I hope that more peaceful time will soon come and that the scientific exchange will no longer be obstructed*". While on Thiem's letter of 28 February 1940, Meinzer wrote as translation: "*We all hope that the light of peace will come to Europe from America. Then I will actually make my trip to America which I have had to give up.*"

It follows from the letters that one or more letters are probably missing and that there might have been correspondence before the first letter of Thiem to Meinzer of 1 December 1936. In this 'first' letter (Fig. 11), Thiem wrote, as translated by Meinzer: "*So you have returned safely to America with your esteemed wife! You have seen the birthplace of your parents and have said to yourself how much has occurred since your parents emigrated to the present time. I am glad that you took back with you good impressions of your European journey. You will certainly think back over it often. Mother Europe is indeed very beautiful, but she is also very tired, if one may be permitted to say so. Your country on the contrary is young and full of development possibilities.*" Thiem further wrote that he was sorry that he could not travel to Edinburgh (for the 1936 International Union of Geodesy and Geophysics (IUGG) General Assembly), as he had hardly any money. Thiem noted that Meinzer travelled to Nancy to see Thiem's good old friend Prof. Imbeaux, a former president of the Commission on Subterranean Water of the IASH-IUGG. Thiem thanked Meinzer that he would send some additional copies of Water-Supply Paper 679-A (i.e. Wenzel, 1936). He also wrote: "*Recently I made the acquaintance of the men in the American Institute in Berlin. They were very friendly and lovable, and my wife had to see the institute. These gentlemen also want to get me some copies of this paper. The demand for it is great, especially from many geological institutions in Germany that are not able to send money because of governmental restrictions.*" In closing the letter, Thiem remarked: "*Please tell your esteemed wife many heart greetings from me and my wife. It was a fine afternoon when you took tea with us. Many thanks for the journey[1] photos. I find them excellent and they will be for me a dear reminder. May you keep real well, and have a happy Xmas; and don't forget your old professional comrade, who greets you many times.*"

---

[1] Here Mr Meinzer makes an (understandable) translation error; the original in German reads 'reizenden', which means 'lovely', however Meinzer confuses it with 'reisen', which means 'to travel'.


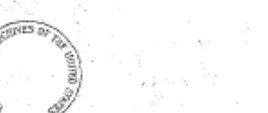

Hydrologisches Büro
**Dr.-Ing. G. Thiem**
Stadtrat a. D.
Beratender Ingenieur

Wasserversorgung
Wasseraufbereitung
Abwasserbeseitigung
Abwasserklärung
Wasseruntersuchung

Leipzig C 1, den   1.Dez.1936
Helfferichstraße 9
Fernsprecher 41582

Herrn

O.E. Meinzer,

*My dear Mr. Meinzer: So you have returned safely to America with your esteemed wife. You have seen the birthplace of your parents, and have said to yourself how much has occurred since your parents emigrated to the present time. I am glad that you took back with you good impressions of your European journey. You will certainly think back over it often. Mother Europe is indeed very beautiful, but she is also very tired, if one may say so. Your country on the contrary is young and full of development possibilities.*

*I was very sorry that I could not journey to Edinburgh, but I had hardly sufficient money. You saw Prof. Jmbeaux in Nancy, he is to me an old and dear friend, to whom I have been greatly attached, and he has a great fund of knowledge.*

*I thank you very much that you will send me some additional copies of W.S.P. 679 A. Recently I made the acquaintance of the men in the American Institute in Berlin. They were very friendly and*

2923 South Dakota Avenue NE
W a s h i n g t o n   USA
- - - - - - - - - - - - - -

Mein Zeichen: J/2

Mein lieber Herr Meinzer !

Sie sind also glücklich mit Ihrer Frau Gemahlin
nach Amerika zurückgekehrt ! Sie haben die Heimat Ihrer Eltern
gesehen und werden sich gesagt haben, was sich alles inzwi-
schen ereignet hat zwischen der Auswanderung ihrer Eltern
und der heutigen Zeit. Es freut mich, dass Sie gute und nach-
haltige Eindrücke von Ihrer Europa-Reise mit nach Haus genom-
men haben; Sie werden sicherlich oft daran zurückdenken. Die
Mutter Europa ist wohl sehr schön, sie ist aber auch sehr müde,
wenn man so sagen darf; Ihr Land hingegen ist ein junges Land
voller Entwicklungsmöglichkeiten.

Ich Es war sehr traurig, dass ich nicht nach
Edinbourgh fahren konnten, doch hätte ich kaum Geld gehabt.
Sie haben den Herrn Prof.Jmbeaux in Nancy gesehen; er ist mir
ein alter lieber Freund, an dem ich sehr gehangen habe und er
verfügt über ein grosses Wissen.


**Figure 11: First page of typewritten letter by Günther Thiem with hand-written translation by Oscar Meinzer from December 1,**
**1936 (Thiem and Meinzer, 1936-1940).**




Oscar Edward Meinzer was born November 28, 1876, on a farm near Davis, Illinois (Sayre, 1948, 1949b). He was one of six
children of William and Mary Julia Meinzer, born in Karlsruhe, Germany. His grandparents and parents emigrated to escape
a culture, which they considered oppressive. "*This may have directly influenced Meinzer's future religious convictions,*
*independent thought, hatred of war, and industriousness*." (Reuss, 2000). The European travel of Meinzer took place in 1936.
He travelled to the IUGG Assembly at Edinburgh, Scotland, but he also visited hydrologists in Germany, Holland and France
(Meinzer, 1936: a 4-page trip report, however it was not published and we have not been able to obtain a copy; Waring and
Meinzer, 1947; Sayre, 1949a). In the interview of C.V. Theis, CV must have been confused about Meinzer bringing back from
this trip the idea of doing a Thiem method pumping test, as the pumping test was executed in 1931 and as there is no indication
that Meinzer made an earlier trip to Europe than 1936 (Sayre, 1949a). Meinzer was the first chairman (1930) of the Hydrology
Section of the American Geophysical Union (AGU), which served as the American National Committee of the IUGG (Meinzer,
1931). He was also from 1936-1948 president of the Commission on Subterranean Water of the IASH-IUGG, from 1947-1948
president of the AGU and as such, he was active in the organization of the IUGG 1936, 1939 and 1948 Assemblies. He
anticipated a second Europe trip to attend the Oslo 1948 IUGG meeting before he passed away (Sayre, 1949a).

Most of the correspondence of Thiem and Meinzer between April 23, 1938, and August 23, 1940, related to a possible
participation of Thiem and a contribution to the IUGG 7[th] Assembly, Washington D.C, September 4-15, 1939. Thiem asked
Meinzer for an invitation to participate in the conference, as normally, these invitations only went to the official institutes and
not to independent hydrological scientists like him. Thiem also expressed his concern if the German government would provide
him with the necessary foreign currency. Meinzer replied that he is happy to note that Thiem and his wife are definitely
planning to come to the US, "*We will do all that we can to make your visit pleasant and profitable*" and "*As you know, Mr*
*Wenzel has done a large amount of work on different methods of determining permeability and flow of ground water so that*
*your contact with him will be mutually helpful.*" He sends a copy of this letter to Prof. Frolow and Dr. Fleming, the latter
General Secretary of the American Geophysical Union and organizer of IUGG 1939 Assembly, and adds a message to Dr
Fleming: "*Dr. Thiem indicates his intention to come to the Washington meeting and to bring his wife with him, provided he*
*can make the necessary arrangements with the German government. It is obvious to me that he does not stand in very well*
*with the official representatives of Germany but we in this country esteem him very highly.*" Meinzer asked Thiem to contribute
to Question No. 3 of the International Commission on Subterranean Water: 'Determination of runoff and physical conditions
of the flow of underground water in natural or altered ground, the flow being natural or induced' of the forthcoming meeting.
This question was coordinated by Leland Wenzel of the USGS. Thiem submitted via the official channel of Dr. Koehne of the
'Landesanstalt für Gewässerkunde' in Berlin (Koehne, 1939) his written contribution "Berechnete und beobachtete
Grundwassermengen" (Thiem, 1939c, 1940d). Meinzer wrote Thiem June 29, 1939: "*Your paper on Question No. 3 with*
*introduction by Dr. Koehne was received a long time ago and is being pre-published for the Washington meeting. Mr. Wenzel*
*and I have read it in part and he will include it in his general report. We find it very interesting.*"





July 31, 1939, Thiem reported about his suffering for weeks: "*My health has not yet fully improved, for I am suffering in my*
*right knee from rheumatism of the joints so that I cannot bear much weight on it. Also I have trouble going up stairs. [...]You*
*cannot imagine how much my refusal (of your invitation) distresses me.*" Meinzer replied: "*I regret very much that the*
*condition of your health will prevent your attending and taking part in the meetings of the Union. As you know, I had*
*anticipated with pleasure meeting you again and discussing with you personally hydrologic problems of mutual interest.*" He
also noted that he translated Thiem's Assembly paper into English for use at the meeting.

On September 18, 1939, three days after the meeting, Meinzer reported to Thiem: "*…although most of the European delegates*
*were not able to attend the meeting in Washington, a considerable number of representative delegates from different countries*
*were nevertheless able to attend and the meeting was very successful. In the Commission on Subterranean Water a total of 55*
*papers were in hand in either printed or typewritten form, and these were effectively reviewed by the general reporters. The*
*relatively few authors who were present were called upon to present their own papers at greater length. The only one of the*
*officers of the Association who was able to attend was Vice-President Slettenmark who served efficiently as the President*
*during the meetings. President Lutschg's Presidential address, which was submitted in German, was translated and presented*
*by Mr. Slettenmark in the English language. It was accompanied by beautiful lantern slides. We all regretted that you and the*
*other German delegates were not able to attend.*" Wenzel (1939) provided a summary on the contributions of Question 3,
while Meinzer (1939) reported on Question no. 2: 'Definitions of the different kinds of subterranean water'. Official reports
of the Assembly, which took place under the emerging clouds of WWII, are provided in Chapman (1939) and Fleming (1940);
"*On August 30, when the European political crises was at its height, it was decided… that the Assembly should be held as*
*scheduled but that its activities should be confined to scientific matters only*". The IUGG President la Cour closed the Assembly
with the words "*…it has been an extremely important meeting, furthering our science and showing to the world a battlefield*
*where only victory can be recorded because even the overthrow of a theory is a victory for truth*" (Fleming, 1940).

On Jan 6, 1940, Thiem wrote to Meinzer that he received a package with extensive documents of the meeting in Washington
and that he now he really regretted that he could not participate. He also noted that he translated into German the Question 3
report of Wenzel (1939) and will publish it in a German professional journal, which he indeed did (Wenzel, 1940). He
continued: "*It is for me a special recognition that the Thiem method for the estimation of the hydraulic conductivity of the*
*subsurface and its water discharge in your country is applied. Do you think, that it later would be suitable to present myself*
*in America to undertake there hydrological investigations for groundwater supply for cities based on my method? I would be*
*very willing to come to America. I would like to ask you to tell me to whom I should direct myself in this case or do you think*
*that your office could take on the negotiation for my appointment as expert? However, these questions can only be discussed*
*with successful prospect when normal times in Europe, let alone in the world, have set in again.*" On February 1, 1940, Meinzer
replied to Thiem that he would like to have a copy of the translated report, and "*We would be glad to have a visit from you at*
*any time. However, I would not wish to encourage you as to the prospects of obtaining professional work in this country. You*





*might be able to make a success of such an undertaking but there are so many difficulties in establishing oneself in a new*
*country that I do not feel at all sure as to the success that you might have.*" Meinzer was friendly, but he definitely discouraged
Thiem from working in the US.

The last letter in the correspondence is from Thiem to Meinzer dated August 23, 1940. Meinzer translated the following lines:
"*Your friendly letter of April 17 was received by me on Aug. 20 … I suppose you will not receive my letter till Christmas.*
*Therefore I will already today wish you a merry Christmas. My wife and I send our best greetings to you and your wife. Auf*
*Wiedersehen either in America of Europe. Yours Dr Engineer G. Thiem.*" It is not known if the correspondence ceased or
continued during or after WWII. However, in 1946 shortly after WWII, Meinzer retired as Geologist in Charge of the Division
of Ground Water. He died June 14, 1948, rather suddenly while taking an afternoon nap, aged 71 (Sayre, 1948).

In the 1949 address book of Leipzig, Thiem is listed as "Beratender Ingenieur für Wasser und Abwasser, Stadtrat a.D."
(Consulting engineer for water and waste water, formerly member of city council), still living in Hillerstraße 9. According to
Grahmann (1960), Günther Thiem was active until his death in Leipzig on August 31, 1959, aged 83.
**5 Conclusions**
Forever, the name Thiem will be connected to the Dupuit-Thiem equation, the first practical model for pump test analysis.
However, father and son Thiem were far more prolific contributors to the canon of methods currently used in hydrogeology
than most people know. All of their method development was done out of practical need, which arose during their many
projects while devising solutions for the many problems they were facing building water supply schemes from scratch. This is
even more remarkable since it was done besides running a successful consulting business and planning many water supply
schemes all over Europe, which today can be found in Germany, Poland, the Czech Republic, Austria, Switzerland, France,
Finland, Sweden, Latvia, Romania, Ukraine and Russia (Fig. 12). The infrastructure they planned and designed is a lasting
legacy since some of their water works are still active today after often more than 100 years, albeit in modernized form (Fig.
13, 14). A few buildings have been preserved as protected monuments, e.g. in Leipzig , Suceava. The most striking buildings
are, of course, the water towers, e.g. in Leipzig (Probstheida, Möckern, Großzschocher), Markranstädt (1895), Liebertwolkwitz
(1904, now used for housing), Olesnica (1898, then Oels), and Strasbourg (1878, now a museum of voodoo).







Figure 12: Map of water works planned and designed by Adolf and Günther Thiem.

While most of the Thiem methods, such as isopotential maps, tracer tests and screened vertical wells were devised by Adolf Thiem, who was a true explorer and inventor, it was Günther´s role to perfect and propagate them, even in the turmoils of two world wars and several regime changes. Considering the cumbersome communication channels of the late 19th and early 20th century and the language barriers of that time, it is amazing to see that both Thiems were in close contact with many leading scientists from Europe and abroad. The field was small, and the members were well aware of the work of each other; publications in different languages did not seem to be a barrier. Especially Günther´s contacts to Oscar Meinzer of the USGS led to the introduction of their methods into the repertoire of English-speaking hydrogeologists. Meinzer's international contacts and his (German) language skills have played a crucial role in the exchange of the strongly developing science of groundwater hydrology.




871

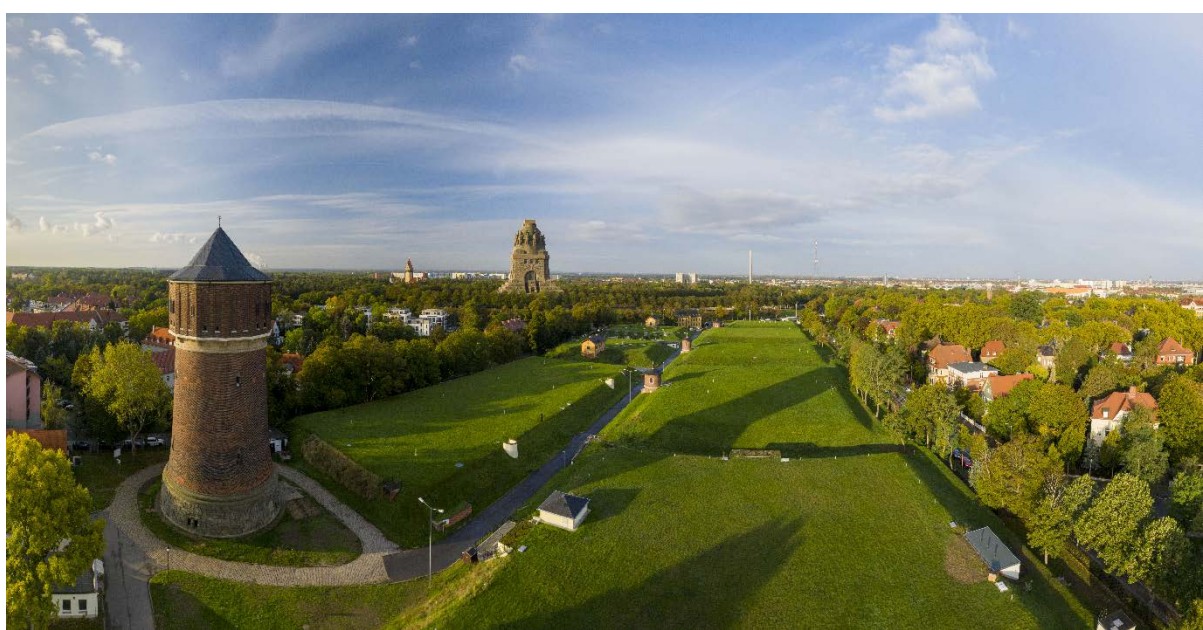

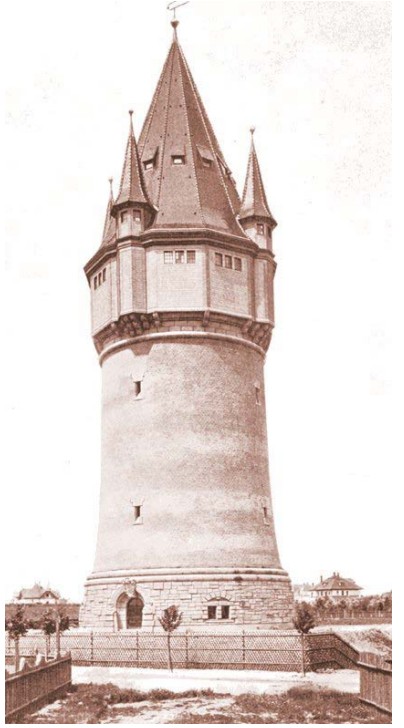

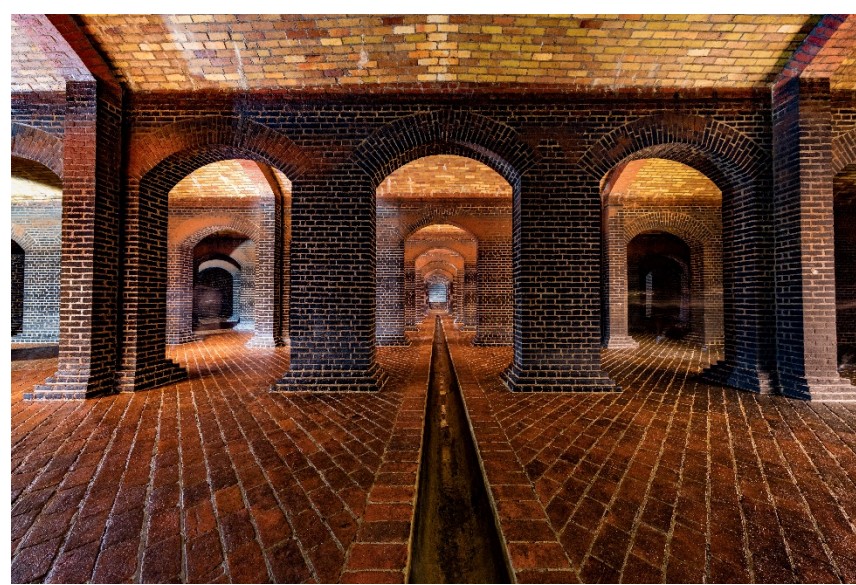

**Figure 13: The Probstheida water works, Leipzig, planned and designed by Adolf Thiem: (above) aerial view with the water tower (foreground left) and water storage cellars with grass cover, the tower of the Völkerschlacht Monument is visible in the background, the small tower to its left is part of the chapel of the Südfriedhof where the Thiem family grave is located, (lower left) water tower in its original shape, around 1907, the roof was damaged in WW II and rebuilt in simplified forms, (lower right): inside of water storage cellar (Photos: Leipziger Gruppe, with permission).**




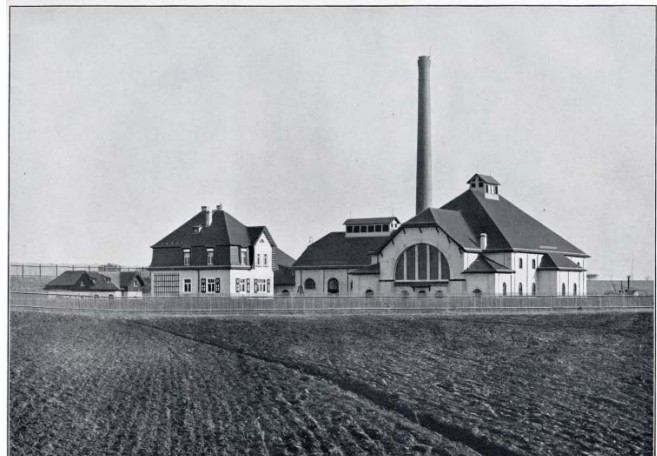
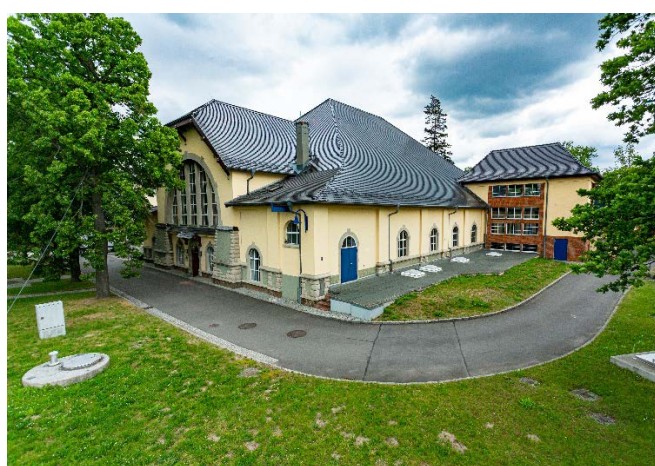
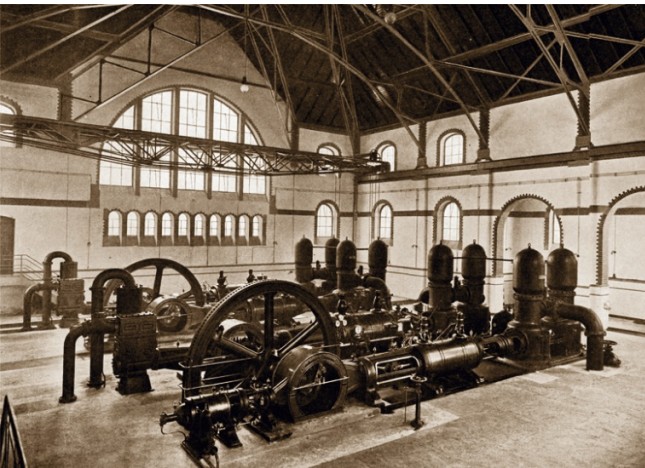
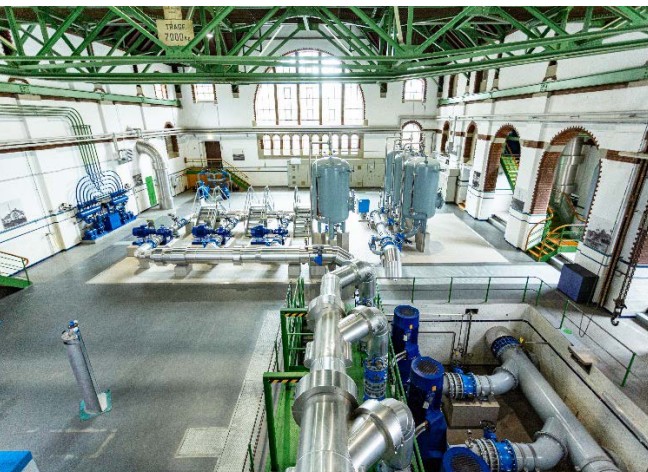

**Figure 14: Buildings and pumps of the Canitz water works then and now, Leipzig: (left) as planned and designed by Adolf and**
**Günther Thiem, status 1912, (right): status today (Photos: Leipziger Gruppe, with permission).**

Both Adolf and Günther Thiem were highly concerned with the practical applicability of their theoretical work and with
presenting it in a way that non-experts could follow their argumentations. In his study for the water supply of Riga, Adolf
Thiem stated that "*Es war mir nicht darum zu tun, Behauptungen und Schlüsse lediglich vom Standpunkt de Fachmannes*
*aufzustellen, sondern ich beabsichtige vielmehr, auch dem außerhalb des Fachs stehenden Leser den logischen Gang der*
*Untersuchungen klarzulegen und ihn so in die Lage zu versetzen, meine Methode kritisch prüfen zu können. (It was not my*
*intention to present my claims and conclusions solely from the point of view of an expert, but to clearly show to a reader, who*
*is not from the field, the logical structure of my investigations, enabling him to critically judge my method)*" (Thiem 1883b).



The engineering work of the Thiems can only be understood in the light of the social and technical problems arising during
the late 19th and the early 20th century. Increasing population, industrialization and urbanization had increased the water
demand but – at the same time – had negatively affected water quality. Groundwater came into focus as a safe, reliable and
often abundant resource to overcome both the demand for a sufficient quantity of water and for improved hygiene by better
water quality. However, little was known about this mysterious underground resource. The Thiems reacted to this societal
problem by adaption of current technology but also by innovation, e.g. the development of new techniques and methods. One
example is the vertical well, which design they improved continuously over several decades, paving the way towards the
modern-day wells. At the same time, they were early adopters of new technology (e.g. the pumps driven by steam engines
used in pumping tests) and new, mass-produced materials (e.g. steel and copper used for wells). Both Thiems were also great
educators and their wealth of publications and presentations shows their tireless dedication to the improvement of water supply.
Hence, the engineering work of the Thiems was in response to the rapidly changing times in which they were living. However,
equally, they benefitted strongly from the developing engineering profession and approaches, providing opportunities for
experimenting and creating solutions for societal problems.

The lives and work of Adolf and Günther Thiem are not only documented in their legacy of references, of which we have tried
to collect and list as many as possible. Several museums hold collections containing reports, letters and photographs. These
include: the archives of the Deutsches Museum (https://www.deutsches-museum.de/en/library/searches/), the Sächsisches
Staatsarchiv (Saxonian State Archive), Dresden and the Museum der Leipziger Stadtgeschichte (Museum of City History),
Leipzig.
**Acknowledgements**
The authors would like to thank to Herfried Apel, a grandson of Adolf Thiem´s oldest son Paul Adolf Thiem, for providing
access to important information on the Thiem family, including the family tree. We also express our gratitude to the Leipziger
Gruppe for providing photographs of the buildings shown in Figure 13 and 14.

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
