# Peer review of "The Thiem team – Adolf and Günther Thiem, two forefathers of hydrogeology"

_Hydrology and Earth System Sciences, 2021_

## Author Response (AR1)

**Response to editor comments on HESS-2021-427**

**The Thiem team – Adolf and Günther Thiem, two forefathers of hydrogeology**

**Georg J. Houben, Okke Batelaan**

**Comments to the author**:
This is an interesting paper. It provides a detailed overview of lives and works of the two Thiems. As such, I do think the text can be published - but only once a few issues have been resolved. These issues relate generally and most importantly to the balance and construction of the historical argument.

We thank the editor for his encouraging words!

Let me start with a detail. In the abstract, it is suggested that it is "remarkable" that the Thiems did their theoretical work while not in university. I would argue that this is an odd statement, as not that many engineers at the time worked at universities, but many did discuss all kinds of issues in the scientific/professional journals. This may seem a detail indeed, but it may represent two issues that needs to be addressed.

We still think that this is "remarkable" for two reasons. (1) While it is true that most forefathers of hydrogeology did not work at universities (or at academies), it was (and still is) a challenge for anyone not working in academia to find the time to write and publish papers. And the Thiems wrote a lot of papers. (2) Today, most hydrogeological research is done at universities, and we take this for granted (some people there even look down on the poor colleagues toiling in the consulting companies). We wanted to show that this was not always so and the Thiems are the best example for this.

Issue 1 relates to the more general context of the Thiems, as engineers and theorists. They were not alone at the time, but we do not really get an idea on that from the paper. The two Thiems lived in different times, did that matter for their work? I would strongly suggest that some ideas on engineering professionalization need to be included, to provide a more general context to understand the Thiems in their time. If needed, I can provide a few publications that are worth using.

We added from line 898 till 918 a text providing more context with respect to the above remarks. We also supported this with a number of references.

Issue 2 relates to the general claim the authors make. Apparently, the Thiems made clear and important contributions - line 23 claims they "laid the foundations" for much. In the text, however, we either find that they modified existing things, or we read in rather speculative terms that "it is very likely" (line 362) or "probable" (line 369). This is one example that for me shows that the fame of the Thiems is not that clear at all. I am sure they worked on much and left many texts, but that in itself is not enough evidence.

Indeed we state that the Thiems "laid the foundations". We have shown and extensively supported this claim, as stated for pumping test analysis, tracer tests, well construction and isopotential maps. Unfortunately, we must remain somewhat cautious for the example discussed in Lines 362 and 369! While it is very likely that the value for the critical entrance velocity goes back to A. Thiem, who was the first to mention it and to assign it a numerical value, there is no direct reference to relating this value to Thiem in the American literature. While we could show conclusively that the Americans did read the Thiems´ work (and were thus probably also aware of the entrance velocity issue), there is no direct reference to Thiem when Bennison discusses it. He might have taken it from Thiem, from a

secondary source or even developed the concept independently. Since Bennison did not state his source, we cannot verify Thiem as the source beyond any doubt.

The importance of the work of the Thiems is now indicated by a google scholar citation analysis added in lines 927-939.

These issues are not that difficult to solve, I would think. After all, this text does not need any speculative claim at all to show that the Thiems are interesting persons, who have done much. They do appear in recent text books (please also include these in the Intro, when this statement is already made), but who were they? That is an interesting read in itself. The authors should check their own text for too optimistic claims, but once this is done, there is enough to enjoy!

Ok, textbook references now also given in introduction.

A few last remarks complete my suggestions:

Why do we read so many direct quotes, with many from the archives, between lines 692-841? It makes the text as a whole unbalanced, and creates the question why these letters are so important? I would recommend explaining this better or rewrite part of the text.

We uphold that the letters are an important piece of the puzzle, since they represent a source that before was simply unknown and never published and investigated. They give unique insights into the scientific and personal interactions of two eminent hydrologist from that time, G. Thiem and O. Meinzer. Direct quotes are a common tool in biographical studies. Not only do they help to avoid an over-interpretation by the biographers, they also show the thinking of the persons involved in their own words.

In any case, all direct quotes need to have a full reference. Archival material needs to be provided in the references too, and needs to have citations in the text too. In the historical science, it is practice to list archival material in a separate list, and cite in foot notes. Archival material, after all, is not a regular secondary source. What we read in lines 727-732 needs to be in the archival list. If I can assist in how to do that, please let me know.

The text in lines 728-734 was shortened by removing the information which is in the reference. However, this text is also meant to show the extent of the communication between Thiem and Meinzer, hence we would like to leave that part in the text. We have adapted the referencing of the archival material and followed the APA style "Collection of letters from an archive" https://apastyle.apa.org/style-grammar-guidelines/references/archival. The collection is now referenced as (USGS, 1936-1940), while specific letters from the collection are cited according the guidelines of the APA style as e.g. (USGS, 1936-1940, Thiem to Meinzer, 1 December, 1936).

Please do not use language like Thiem "instinctively" understanding something (line 364). I would like to see that such remarks are all removed.

OK, removed as requested.

All in all, there is still some work to be done on this paper, but as I think it is quite doable, and the text as such is rather interesting and does not need too much speculation, I decided that these changes would qualify for "minor revisions". I look forward to the revised version.